# Shapley Values of Structured Additive Regression Models and Application to RKHS Weightings of Functions

**Gabriel Dubé**  *gadub44@ulaval.ca*
*Département d'informatique et de génie logiciel*
*Université Laval*

**Mario Marchand**  *mario.marchand@ift.ulaval.ca*
*Département d'informatique et de génie logiciel*
*Université Laval*

**Reviewed on OpenReview:** *https: // openreview. net/ forum? id= aWRMvXTvPf*

## Abstract

Shapley values are widely used in machine learning to interpret model predictions. However, they have an important drawback in their computational time, which is exponential in the number of variables in the data. Recent work has yielded algorithms that can efficiently and exactly calculate the Shapley values of specific model families, such as Decision Trees and Generalized Additive Models (GAMs). Unfortunately, these model families are fairly restricted. Consequently, we present STAR-SHAP, an algorithm for efficiently calculating the Shapley values of Structured Additive Regression (STAR) models, a generalization of GAMs which allow any number of variable interactions. While the computational cost of STAR-SHAP scales exponentially in the size of these interactions, it is independent of the total number of variables. This allows the interpretation of more complex and flexible models. As long as the variable interactions are moderately-sized, the computation of the Shapley values will be fast, even on high-dimensional datasets. Since STAR models with more than pairwise interactions (e.g. GA2Ms) are seldom used in practice, we also present a new class of STAR models built on the RKHS Weightings of Functions paradigm. More precisely, we introduce a new RKHS Weighting instantiation, and show how to transform it and other RKHS Weightings into STAR models. We therefore introduce a new family of STAR models, as well as the means to interpret their outputs in a timely manner.

## 1 Introduction

The ability to interpret a machine learning model is an increasingly important consideration. Highly complex models can achieve great accuracy, but understanding their inner workings is challenging. Why did the model offer this particular prediction for this input? The decision was based on which features of the data? On the other hand, simpler models, like Generalized Additive Models (GAMs[1], Hastie (2017)) or Decision Trees (Kotsiantis, 2013), are inherently easier to understand, but do not always reach the same levels of performance (though recent work appears to show otherwise (Rudin et al., 2024)). Work to improve explanation methods, especially the interpretation of more complex models, can help bridge this gap.

Informally, we mean by *interpreting a model* the task of explaining its output. In sensitive contexts such as healthcare (Caruana et al., 2015), energy systems (Zhang & Chen, 2024) or machine fault diagnosis (Brusa et al., 2023), to give only a few examples, mistakes can be costly, and so the ability to understand, and thus trust, an AI model is crucial. In this paper, we specifically consider the **attribution problem**, which consists in distributing the prediction score between the data features (Sundararajan & Najmi, 2020). On one

---

[1]Table 1 lists all the acronyms used in this paper.

end of the spectrum of interpretability lie linear models, where the contribution of each variable is obvious and independent of all other variables. On the other end, we have large deep neural networks, the incredibly complex inner workings of which are singularly difficult to elucidate (Montavon et al., 2018).

Post-hoc interpretability methods attempt to explain even complex models through various means. Just some examples are LIME (Ribeiro et al., 2016), which explains the prediction of a complex model by approximating the model locally around the instance using a simple predictor, and Partial Dependance Plots (Friedman, 2001), which visualize the effect of a variable by holding others constant.

Another widely used tool for interpreting models, and the subject of this paper, is the Shapley value (see e.g. Winter (2002)), originally devised in the context of game theory (Shapley et al., 1953). Adapted to the machine or statistical learning context, it can be used to assess the contribution of each variable to the output of a model, for a given example. Additionally, it can be averaged over a dataset to see the relative importance of each variable.

With all of this in mind, the purpose of this paper is twofold. First, we present an algorithm for efficiently calculating the Shapley values of a large family of prediction models, Structured Additive Regression (STAR) models. STAR models are a further generalization of GAMs that allow any number of variable interactions in each term, making them potentially much more expressive models. The algorithm that we present does have an exponential complexity in the *size* of these interactions, but has no explicit dependency on the total number of variables. This allows interpreting models with many more variable interactions, helping to make complex models more viable in sensitive domains. We show that our algorithm is especially useful in the context of high-dimensional datasets, such as Wood et al. (2015), because of this lack of an explicit dependency on the total number of variables.

Second, we improve the understanding and usability of a newly introduced class of models, the RKHS Weightings of Functions (Dubé & Marchand, 2024), mainly in the context of STAR models. To do so, we present a new RKHS Weighting instantiation (Dubé & Marchand, 2024) which is better suited to regression. Then, we show how to transform it and other RKHS Weightings into STAR models. This yields an entirely new family of STAR models. We hope that these developments will foster interest and further research into RKHS Weightings, which we believe have significant untapped potential.

The rest of the paper is as follows. Section 2 covers Shapley values and RKHS Weightings of Functions. Section 3 presents the formula and corresponding algorithm for calculating the Shapley values of a STAR model. Section 4 presents a new RKHS Weighting instantiation, and shows how to transform an RKHS Weighting model into a STAR model. Section 5 contains various experiments, showing the computation time of Shapley values using our new algorithm, the regression and time series performance of the models introduced in this paper, and an example of interpreting a model using our new algorithm. Finally, in Section 6, we go over some limitations of this paper as well as future work to be done, and conclude in Section 7.

## 2 Review of Shapley values and RKHS Weightings

The two main subjects of this paper are Shapley values and RKHS Weightings, two tools we use within the context of machine learning. We start by defining the basic machine learning notions that we use in this paper. Then, we present our definition of the Shapley value, and end on a summary of RKHS Weightings.

### 2.1 Machine learning

The **regression** setting in machine learning consists of learning to predict real values from examples. Given an **instance space** $\mathcal{X}$, usually $\mathcal{X} = \mathbb{R}^n$, the **label space** $\mathcal{Y} = \mathbb{R}$, and a **dataset** of examples $\mathcal{S} = \{(x_i, y_i)\}_{i=1}^m \subset \mathcal{X} \times \mathcal{Y}$, our task is to find a **predictor** (also called **model**) $h : \mathcal{X} \to \mathcal{Y}$ that optimizes some objective function of the sample data. To measure the quality of a model, we use a **loss function** $\ell : \mathcal{Y} \times \mathcal{Y} \to \mathbb{R}$, which in this paper is the **squared loss**:

$$\ell(h(x), y) := \frac{1}{2}(h(x) - y)^2. \tag{1}$$

| Acronym | Meaning | Relevant link |
|---|---|---|
| GAM | Generalized Additive Model | Hastie (2017) |
| GA2M | GAM with pairwise interactions | Lou et al. (2013) |
| STAR (model) | Structured Additive Regression (model) | Equation (16) |
| $k$-STAR model | STAR model using only $k$-way interactions | |
| RKHS | Reproducing Kernel Hilbert Space | Berlinet & Thomas-Agnan (2011) |
| MSE | Mean Square Error | Equation (2) |
| $R^2$ | Coefficient of determination | Equation (3) |
| SHAP | SHapley Additive exPlanations | Lundberg & Lee (2017) |
| SHAP-IQ | SHAPley Interaction Quantification | Fumagalli et al. (2023) |
| STAR-SHAP | Structured Additive Regression SHAP | Algorithm 1 |
| erf | Error function, $\mathrm{erf}(x) := \frac{2}{\sqrt{\pi}} \int_0^x e^{-t^2} \, \mathrm{d}t$ | |
| ReLU | Rectified Linear Unit, $\mathrm{ReLU}(z) := \max(0, z)$ | |
| RWSign | RKHS Weighting using sign function as base predictor | I1 in Dubé & Marchand (2024) |
| RWStumps | RKHS Weighting using decisions stumps as base predictor | I2 in Dubé & Marchand (2024) |
| RWReLU | RKHS Weighting using ReLU as base predictor | Section 4.1 |
| EBM | Explainable Boosting Machine | Lou et al. (2013) |
| SVR | Support Vector Regression | Pedregosa et al. (2011) |

Table 1: Acronyms used in this paper.

The **empirical risk** is the sample average loss. When using the squared loss, this is called the **Mean Square Error** (MSE):

$$\mathrm{MSE}_{\mathcal{S}}(h) := \frac{1}{2m} \sum_{i=1}^{m} (h(x_i) - y_i)^2. \tag{2}$$

The MSE has units of the labels squared. To successfully compare experimental results on different datasets, we instead report the **coefficient of determination**:

$$R_{\mathcal{S}}^2(h) := 1 - \frac{\sum_{i=1}^{m}(y_i - h(x_i))^2}{\sum_{i=1}^{m}(y_i - \bar{y})^2}, \tag{3}$$

where $\bar{y} := \frac{1}{m} \sum_{i=1}^{m} y_i$ is the average label.

In practice, simply minimizing the sample MSE is insufficient. We seek a model that will successfully generalize to new, yet-to-be-seen instances. Indeed, we can assume that the data comes from some unknown distribution $\mathcal{D}$ over $\mathcal{X} \times \mathcal{Y}$. The true risk, or true MSE, is the expected squared loss over this distribution:

$$\mathrm{MSE}_{\mathcal{D}}(h) := \frac{1}{2} \mathop{\mathbb{E}}_{(x,y) \sim \mathcal{D}} \left[ (h(x) - y)^2 \right]. \tag{4}$$

We estimate this value by calculating the MSE on a set of examples distinct from the sample.

A **learning algorithm** $\mathcal{A}$ is a function that takes in a sample $\mathcal{S}$, and returns a predictor, i.e. $\mathcal{A}(\mathcal{S}) = h(x)$. It is this algorithm that seeks to minimize the true risk by optimizing a function of the best available proxy, the empirical MSE. Most learning algorithms have **hyperparameters**; the number of trees in a random forest; the learning rate of a neural network; the regularization parameter of the kernel ridge regression. Each combination of parameters yields an algorithm. Suppose we have $C$ combinations to choose from. This implies we have the candidate algorithms $\mathcal{A}_1$ to $\mathcal{A}_C$. To select the best one, we use **cross-validation**, more specifically $k$-fold cross-validation. The sample $\mathcal{S}$ is first divided into $k$ equal disjoint subsets $\mathcal{S}_1$ to $\mathcal{S}_k$. Each algorithm $\mathcal{A}_c$ is applied $k$ times, giving us the $k$ models $\mathcal{A}_c(\mathcal{S} \setminus \mathcal{S}_1), \ldots, \mathcal{A}_c(\mathcal{S} \setminus \mathcal{S}_k)$. We can estimate the true risk of each of these models through its MSE on the holdout set, e.g. $\mathrm{MSE}_{\mathcal{D}}(\mathcal{A}_c(\mathcal{S} \setminus \mathcal{S}_1)) \approx \mathrm{MSE}_{\mathcal{S}_1}(\mathcal{A}_c(\mathcal{S} \setminus \mathcal{S}_1))$. In turn, we can estimate the true risk of $\mathcal{A}_c(\mathcal{S})$ (the model trained on the entire dataset) as the average MSE of these $k$ models:

$$\mathrm{MSE}_{\mathcal{D}}(\mathcal{A}_c(\mathcal{S})) \approx \frac{1}{k} \sum_{i=1}^{k} \mathrm{MSE}_{\mathcal{S}_i}(\mathcal{A}_c(\mathcal{S} \setminus \mathcal{S}_i)). \tag{5}$$

The algorithm with the lowest approximated MSE is chosen, and trained again on the entire sample.

## 2.2 Shapley values

The Shapley value was originally derived by Shapley et al. (1953) as a uniquely interesting attribution method in the context of a game with $n$ players. Denoting $[n] := \{1, \ldots, n\}$ the set of natural numbers up to $n$, and $\mathcal{P}([n])$ its power set, the Shapley value revolves around a **value function** $\nu : \mathcal{P}([n]) \to \mathbb{R}$. Given a coalition of players $I \subseteq [n]$, $\nu$ is the payoff that this coalition can achieve in the game by playing together. The Shapley value of player $i$ is defined as:

$$\phi_i(\nu) = \sum_{I \subseteq [n] \setminus \{i\}} \frac{|I|! \, (n - |I| - 1)!}{n!} \left(\nu(I \cup \{i\}) - \nu(I)\right). \tag{6}$$

It is the average value added by player $i$ when it joins a coalition where $i$ is initially absent. The Shapley value is the unique attribution method that satisfies four key properties (Fryer et al., 2021):

1. **Efficiency.** The Shapley values of all players sum to the value of all players together:

$$\sum_{i=1}^{n} \phi_i(\nu) = \nu([n]). \tag{7}$$

   In simpler words, the gain is distributed among all the agents/players.

2. **Symmetry.** Two players with equal contributions have equal Shapley values:

$$\left(\forall I \subseteq [n] \setminus \{i, j\}, \nu(I \cup \{i\}) = \nu(I \cup \{j\})\right) \implies \phi_i(\nu) = \phi_j(\nu). \tag{8}$$

3. **Null player.** A player that has no contribution has a Shapley value of 0:

$$\left(\forall I \subseteq [n] \setminus \{i\}, \nu(I \cup \{i\}) = \nu(I)\right) \implies \phi_i(\nu) = 0. \tag{9}$$

4. **Linearity.** The Shapley values of a sum $\nu_1 + \nu_2$ of value functions are the sum of the Shapley values of the individual functions:

$$\forall i \in [n], \phi_i(\nu_1 + \nu_2) = \phi_i(\nu_1) + \phi_i(\nu_2). \tag{10}$$

These properties make the Shapley value a uniquely interesting attribution method for the contribution of each variable to the prediction of a model. Although some raise doubts on its reliability (Huang & Marques-Silva, 2023) or the desirability of these four properties (Fryer et al., 2021), we do not address this debate in this paper.

Different value functions yield different Shapley values, so the choice of value function is crucial. In this paper, we wish to attribute to each variable in a dataset its contribution to the output of a model. Consider the instance space $\mathcal{X} = \mathbb{R}^n$, so that the natural numbers up to $n$, which we denoted $[n] := \{1, \ldots, n\}$, identify the $n$ variables that make up an instance $x = (x_1, \ldots, x_n) \in \mathcal{X}$. For some model $h : \mathcal{X} \to \mathbb{R}$, what is the contribution of variable $i \in [n]$ to the model output $h(x)$? In general, it is not possible to see what a model would output given a coalition $I \subset [n]$ of variables, since most models are undefined with missing values. To sidestep this issue, the solution most commonly used (Lundberg & Lee, 2017) is to intervene on the values of $x$ on the variables outside of the coalition. To that end, we define the **replacement function**. For any feature subset $P \subseteq [n]$, and $x, z \in \mathcal{X}$, the vector $r_P(z, x)$ is $z$, except for the variables in $P$ that are replaced by the values in $x$:

$$r_P(z, x)_i = \begin{cases} z_i & i \notin P \\ x_i & i \in P. \end{cases} \tag{11}$$

Finally, suppose that we have an instance $x$ of interest and a **background** $\mathcal{S} \subseteq \mathcal{X}$ consisting of a set of instances. Denote by $U(\mathcal{S})$ the uniform distribution over $\mathcal{S}$. The value function $\nu : \mathcal{P}([n]) \to \mathbb{R}$ that we focus on is:

$$\nu(I) := \underset{z \sim U(\mathcal{S})}{\mathbb{E}} [h(r_I(z, x))]. \tag{12}$$

In simple words, the output of $h$ is averaged over the background, but the variables in $I$ are fixed to their value in $x$. This value function yields a so-called **interventional** Shapley value, after the act of replacing feature values with those from another instance, i.e. "intervening" on these features. This is the Shapley value calculated by SHAP (Lundberg & Lee, 2017).

Rather than Equation (6), we use an equivalent form using permutations of $[n]$ (see e.g. Equation (3) of Khorrami Chokami & Rabitti (2024)). A **permutation** is a bijective function $\pi : [n] \to [n]$; $\pi(i)$ is the position of variable $i$ in the permutation. Let $\Omega$ be the set of all permutations of $[n]$, and $U(\Omega)$ be the uniform distribution over $\Omega$. Also define:

$$\pi_{:i} := \left\{ j \in [n] \,\middle|\, \pi(j) < \pi(i) \right\}, \tag{13}$$

the set of variables preceding (but excluding $i$) in the order implied by the permutation $\pi$. Given all of these notations, the **Shapley value** of variable $i$, for example $x$, for the predictor $h$, is defined as:

$$\phi_i^{\text{SHAP}}(h, x) := \underset{z \sim U(\mathcal{S})}{\mathbb{E}} \, \underset{\pi \sim U(\Omega)}{\mathbb{E}} \left[ h(r_{\pi_{:i} \cup \{i\}}(z, x)) - h(r_{\pi_{:i}}(z, x)) \right]. \tag{14}$$

(Definition taken from Laberge et al. (2023).)

Calculating Equation (14) directly is in $\mathcal{O}(n!)$, as it requires enumerating every permutation of $[n]$. It therefore cannot reasonably be used except on particularly low-dimensional datasets. However, some model families admit much faster formulas or algorithms for calculating the Shapley values. Such model families include linear models (Lundberg & Lee, 2017), Generalized Additive Models (GAMs), and Trees and ensembles of Trees (Lundberg et al., 2018). The common characteristic of these models is limited variable interactions, so that the contribution of a particular variable can be calculated independently of the variables with which it does not interact. For example, the Shapley values of a linear model $h(x) = \sum_j w_j x_j$ are obtained easily from its coefficients by the formula $\phi_i^{\text{SHAP}}(h, x) = w_i(x_i - \mathbb{E}_{z \sim U(\mathcal{S})} z_i)$ (Lundberg & Lee, 2017); the Shapley values of a GAM $h(x) = \sum_j f_j(x_j)$ are given by:

$$\phi_i^{\text{SHAP}}(h, x) = f_i(x_i) - \underset{z \sim U(\mathcal{S})}{\mathbb{E}} [f_i(z_i)]. \tag{15}$$

(The proof is in the appendix.)

Structured Additive Regression (STAR) models are a generalization of both linear models and GAMs (see e.g. Mayer et al. (2021)). A STAR model allows variable interactions by defining the predictor over subsets of the data features:

$$h(x) = \sum_{I \subseteq [n]} f_I(x_I), \tag{16}$$

where $x_I$ is the partial vector $(x_{I_1}, \ldots, x_{I_{|I|}})$, and $f_I$ is a function defined over the feature subset $I$. Importantly, not all of these functions $f_I$ need to be nonzero. We can indeed choose which variable interactions are desired by selecting which feature subsets $I$ are associated with a nonzero $f_I$. For example, GAMs use only singleton feature subsets, i.e. individual variables, while GA2Ms (Lou et al., 2013) use up to pairwise interactions between the variables.

STAR models have various names in the literature. Bordt & von Luxburg (2023) define Generalized Additive Models of order $k$ as models of the form:

$$h(x) = \sum_{I \subseteq [n], |I| \leq k} f_I(x_I), \tag{17}$$

and introduce $k$-Shapley Values, which attribute a contribution to each feature subset, instead of only individual variables. This generalizes Shapley values, but suffers from the same high computational cost.

Laberge et al. (2024) call Equation (16) a Functional Decomposition, following previous work on the subject (Hooker, 2004; Kuo et al., 2010), and show among other things how to obtain the Shapley values of a function from its Interventional Decomposition (Kuo et al., 2010). This, again, is a high cost procedure.

In short, there are two types of formulas or algorithms for exactly calculating the Shapley values of a model. Model-agnostic formulas like the original Equation (13) of Shapley et al. (1953), Equation 14, Equation (11) of Bordt & von Luxburg (2023); and algorithms like SHAP Lundberg & Lee (2017). These algorithms all suffer from the same delibitating complexity with regard to the total number of variables. On the other hand, model-specific formulas, such as those previously mentioned for linear models and GAMs, and algorithms, like TreeSHAP (Lundberg et al., 2018), are vastly more efficient, but are constrained to their respective model families.

In order to work with higher dimensional data and arbitrary models, model-agnostic approximation algorithms have been developed. Notably among these are KernelSHAP (Lundberg & Lee, 2017), Unbiased KernelSHAP (Covert & Lee, 2020) and SHAP-IQ (Fumagalli et al., 2023). These provide approximative Shapley values (and even $k$-Shapley values in the case of SHAP-IQ) without the exponential complexity in the number of variables. Although the values are approximations, rather than the exact Shapley values, Unbiased KernelSHAP and SHAP-IQ have been shown to be unbiased, i.e. they return values which, in expectation, are the correct ones, and consistent, meaning that the variance of the approximations goes to zero at the limit as the Monte Carlo sampling size increases. Approximation is therefore a valid alternative when efficient model-specific exact formulas or algorithms are not known or do not exist.

In Section 3, we present STAR-SHAP, a model-specific algorithm for calculating the exact Shapley values of any STAR model. As long as the maximal number $k$ of interactions is limited, this is an efficient algorithm. Specifically, its computational complexity does scale exponentially in the maximal size $k$ of the feature subsets and scales linearly in the number of subsets $I$ used in Equation (16), but does not explicitly depend on the dimensionality $n$ of the data. This allows using many more interaction terms, while maintaining the ability to interpret the model predictions through its Shapley values.

In Section 5.2, we compare the computation time of this algorithm to the model-agnostic methods we previously mentioned: SHAP, KernelSHAP, Unbiased KernelSHAP, and SHAP-IQ. We also study the quality of these approximations in relation to the exact values provided by our algorithm.

As a final note to this section, we wish to mention the recent work of Khorrami Chokami & Rabitti (2024), who provides an exact formula for the Shapley values of GAMs, however using a different value function. Where we seek to explain the output of a model on specific instances, Khorrami Chokami & Rabitti (2024) are interested in the variance explained by each variable across the entire data distribution. They therefore use the value function $\nu(I) := \mathrm{Var}_x \left[ \sum_{i \in I} f_i(x_i) \right]$, which yields, as the corresponding Shapley value, a global variable importance index, as opposed to a granular instance-specific explanation of the output of the model. This shows how rich the space of possible Shapley values are when considering all the possible value functions that can be used. This thought aside, we focus on the value function given by Equation (12) for the remainder of this paper.

## 2.3 RKHS Weightings

Dubé & Marchand (2024) introduced a new and flexible model family, parameterized through the tuple $(\mathcal{W}, \phi, \mathcal{K}, p)$ of a parameter space $\mathcal{W}$; base predictor $\phi : \mathcal{W} \times \mathcal{X} \to \mathbb{R}$; kernel $\mathcal{K} : \mathcal{W} \times \mathcal{W} \to \mathbb{R}$; distribution $p$ over $\mathcal{W}$. The kernel $\mathcal{K}$ implicitly defines a **reproducing kernel Hilbert space** (RKHS) $\mathcal{H}$, a space of functions of the form $\alpha : \mathcal{W} \to \mathbb{R}$ with particular properties. (A thorough understanding of reproducing kernel Hilbert spaces is not required for this paper, but the interested reader can see Dubé & Marchand (2024) for a short primer on RKHS.) They define the operator $\Lambda : \mathcal{H} \to L^\infty(\mathcal{X})$, which takes as input a weight function $\alpha \in \mathcal{H}$, and returns the bounded predictor $\Lambda\alpha : \mathcal{X} \to \mathbb{R}$ via the equation:

$$\Lambda\alpha(x) := \mathop{\mathbb{E}}_{w \sim p} [\alpha(w)\phi(w, x)]. \tag{18}$$

In practice, the weight function $\alpha \in \mathcal{H}$ takes the form of a sum $\sum_{t=1}^{T} a_t \mathcal{K}(w_t, \cdot)$ (a property of the RKHS), for some real coefficients $a_1, \ldots, a_T$ and $w_1, \ldots, w_T \in \mathcal{W}$. The model can then be written as:

$$\Lambda\alpha(x) = \sum_{t=1}^{T} a_t \underset{w \sim p}{\mathbb{E}} \left[\mathcal{K}(w_t, w)\phi(w, x)\right]. \tag{19}$$

Therefore, an **RKHS Weighting** model is a weighted sum of the partial functions defined by the expectations $\mathbb{E}_{w \sim p}\left[\mathcal{K}(w_t, w)\phi(w, x)\right]$. Each such function of $x$ is parameterized by a $w_t \in \mathcal{W}$. While this is in theory a highly flexible family of models, the current bottleneck for its utilization is the requirement for exactly calculating these expectations. This can be done, depending on the choice of instantiation $(\mathcal{W}, \phi, \mathcal{K}, p)$. Dubé & Marchand (2024) presented two instantiations of the model, which they called I1 and I2. We will refer to these as RWSign and RWStumps respectively after their base predictor $\phi$, a simple sign function for RWSign, and a decision stump for RWStumps, with RW standing for RKHS Weighting. The reader can refer to Table 3 in Section 4 for the explicit definition of RWSign and RWStumps, and to Table 1 of Dubé & Marchand (2024) for the analytical form of the expectation $\mathbb{E}_{w \sim p}\left[\mathcal{K}(w_t, w)\phi(w, x)\right]$. In both cases, the entire model (Equation (19)) becomes a weighted sum of error functions[2], a highly expressive form.

Since Equation (18) represents an aggregation of base predictor predictions (with regard to a distribution $p$ and a weight function $\alpha$), and since the base predictor, in both cases, can only take two values, we surmise that while these instantiations can be used for regression, they are first of all intended for classification. We introduce in Section 4 a new instantiation which we expect to be better suited to regression, using the **rectified linear unit**[3] (ReLU) as the base predictor. Furthermore, we show in Section 4 how to generally build RKHS Weightings that are also STAR models. We also show specifically how some existing instantiations, which satisfy some conditions, can easily be converted into STAR models. This includes RWSign and RWRelu, the instantiation that we introduce in Section 4. (RWStumps is in fact already a GAM, and so does not require any modification.) These models are all interpretable, as their Shapley values can be calculated using Algorithm 1, which we introduce in the next section.

## 3 Shapley values of STAR models

The following theorem encapsulates the main contribution of this paper, an efficient formula for calculating the Shapley values of Structured Additive Regression (STAR) models.

**Theorem 3.1** (Shapley values of a STAR model)**.** *Consider a STAR model:*

$$h(x) = \sum_{I \subseteq [n]} f_I(x_I). \tag{20}$$

*Consider the replacement function $r_P$ defined in Equation (11), and the sets:*

$$\mathcal{A}^+(i, I) := \{A \subset I \mid i \in A, 1 \leq |A| < |I|\} \tag{21}$$

$$\mathcal{A}^-(i, I) := \{A \subseteq I \mid i \in A, 1 < |A| \leq |I|\}. \tag{22}$$

*Then the Shapley values of the model are:*

$$\phi_i^{\text{SHAP}}(h, x) = \underset{z \sim U(\mathcal{S})}{\mathbb{E}} \left[ \sum_{I : f_I \neq 0, i \in I} \left( \frac{f_I(x_I) - f_I(z_I)}{|I|} + \right. \right.$$

$$\left. \left. \sum_{A \in \mathcal{A}^+(i,I)} \frac{f_I(r_A(z, x)_I)}{|A|\binom{|I|}{|A|}} - \sum_{A \in \mathcal{A}^-(i,I)} \frac{f_I(r_{A \setminus \{i\}}(z, x)_I)}{|A|\binom{|I|}{|A|}} \right) \right]. \tag{23}$$

*Proof.* The proof is a series of combinatorial arguments, and can be found in the appendix. $\qquad\square$

---

[2] $\operatorname{erf}(x) := \frac{2}{\sqrt{\pi}} \int_0^x e^{-t^2} \mathrm{d}t$
[3] $\operatorname{ReLU}(z) := \max(0, z)$

Assuming that the feature subsets $I$ are limited in size, then Equation (23) is vastly more efficient to calculate than the definition of the Shapley values in Equation (14). Instead of going through all the permutations of $[n]$, Equation (23) requires only enumerating the subsets of each feature subset $I$ present in Equation (16). However, we should note that using Equation (23) as it is to calculate all the Shapley values for a given instance leads to many redundant computations. Indeed, observe that:

1. For any $j \in A \in \mathcal{A}^+(i, I)$, we also have $A \in \mathcal{A}^+(j, I)$. This means that the computation $f_I(r_A(z, x)_I)$ appears in the calculation of the Shapley value of each variable in $A$.

2. Consider any $A \in \mathcal{A}^+(i, I)$, and a variable $j \in I \setminus A$. Then $A \cup \{j\} \in \mathcal{A}^-(j, I)$, and so the computation $f_I(r_{(A \cup \{j\}) \setminus \{j\}}(z, x)_I)$ is equal to $f_I(r_A(z, x)_I)$.

In short, when calculating the Shapley values of all variables (rather than a single one), the same computation $f_I(r_A(z, x)_I)$ is done exactly $|I|$ times; $|A|$ times from the first observation, and $|I| - |A|$ times from the second observation. We can save a factor $|I|$ by doing this computation only once. The term $\frac{f_I(x_I) - f_I(z_I)}{|I|}$ can also be calculated once, as the value is the same for all variables. These optimizations in mind, we present Algorithm 1, which simultaneously calculates all the Shapley values of a STAR model for a given instance.

---

**Algorithm 1** STAR-SHAP: Shapley Values of a Structured Additive Regression Model

---

**input** Input space $\mathcal{X} \subseteq \mathbb{R}^n$, instance $x \in \mathcal{X}$, STAR model $h(x) = \sum_{I \subseteq [n]} f_I(x_I)$, sample $\mathcal{S} \in \mathcal{X}^m$

    Initialize $\Phi = (\phi_1, \ldots, \phi_n) = 0 \in \mathbb{R}^n$
    **for** $z \in \mathcal{S}, I \subseteq [n]$ such that $f_I \neq 0$ **do**
        baseline $\leftarrow \frac{f_I(x_I) - f_I(z_I)}{|I|}$
        **for** $i \in I$ **do**
            $\phi_i \leftarrow \phi_i + $ baseline
        **end for**
        **for** $A \subseteq I, A \neq \emptyset$ **do**
            term $\leftarrow f_I(r_A(z, x)_I)$
            **for** $i \in I$ **do**
                **if** $i \in A$ and $A \in \mathcal{A}^+(i, I)$ **then**
                    $\phi_i \leftarrow \phi_i + \frac{\text{term}}{|A|\binom{|I|}{|A|}}$
                **else if** $i \in I \setminus A$ and $(A \cup \{i\}) \in \mathcal{A}^-(i, I)$ **then**
                    $\phi_i \leftarrow \phi_i - \frac{\text{term}}{(|A|+1)\binom{|I|}{|A|+1}}$
                **end if**
            **end for**
        **end for**
    **end for**
**output** $\frac{1}{m}\Phi$

---

A highly interesting characteristic of Algorithm 1 is the lack of an explicit dependency on the dimensionality of the data in its algorithmic complexity. Indeed, the dominant operation is the computation of the partial model outputs $f_I(r_A(z, x)_I)$. Note that, at this point, we make no assumption on the partial models $f_I$ other than that they utilize $|I|$ variables. Let us therefore suppose that some nondecreasing function $F : \mathbb{N} \to [0, \infty)$ characterizes the computational complexity of the partial models, so that the cost of calculating $f_I(x_I)$ is in $\mathcal{O}(F(|I|))$ for all $I$ (and all $x$). Denote $k$ the maximal size of the feature subsets associated with a nonzero partial model ($f_I \neq 0$), and $N$ the number of such feature subsets. Algorithm 1 is a loop over these $N$ feature subsets and over the $m$ examples in the background $\mathcal{S}$. For each feature subset $I$, the algorithm then iterates over all its subsets $A \subseteq I$, calculating the partial output $f_I(r_A(z, x)_I)$ each time. Since there are $2^k$ subsets to a set of size $k$, this dominant operation is done a total of $mN2^k$ times, each with a cost of $\mathcal{O}(F(|I|))$. Since $|I| \leq k$, this leads to a favorable algorithmic complexity of $\mathcal{O}(mN2^kF(k))$, independently of the dimensionality $n$ of the data. One could argue that $N$, the number of feature subsets $I$ defining the

| Algorithm | Target models | Exact? | Complexity |
|---|---|---|---|
| TreeSHAP | Trees, ensembles of trees | Yes | $\mathcal{O}(mTLD^2)$ |
| STAR-SHAP | STAR models | Yes | $\mathcal{O}(mN2^k F(k))$ |
| SHAP | Any | Yes | $\mathcal{O}(m2^n F(n))$ |
| KernelSHAP | Any | No | $\mathcal{O}(Kn^2 + n^3 + KmF(n))$ |
| Unbiased KernelSHAP | Any | No | $\mathcal{O}(Kn^2 + KmF(n))$ |
| SHAP-IQ | Any | No | $\mathcal{O}(Kn^2 + KmF(n))$ |

Table 2: Computational complexity of various Shapley value algorithms for calculating all $n$ Shapley values of a single $n$-dimensional instance. All algorithms: $m$ is the size of the background sample. TreeSHAP: $T$ is the number of trees, $L$ is the number of leaves in a tree, and $D$ is the depth of a tree. STAR-SHAP: $N$ is the number of nonzero terms in the model, $k$ is the maximal number of interactions per term, $F(k)$ is the cost of computing a term with $k$ interactions. SHAP: $n$ is the number of variables, $F(n)$ is the cost of computing the model. All others: $K$ is the sampling size for the approximation. The reported complexities of KernelSHAP, Unbiased KernelSHAP and SHAP-IQ are those of their implementation in Muschalik et al. (2024), version 1.1.0, the software package used for our experiments in Section 5. Note that their implementations of Unbiased KernelSHAP and SHAP-IQ coincide for calculating 1-Shapley values, the use case for this paper. This is faster than the original Unbiased KernelSHAP implementation, while yielding the same values (Fumagalli et al., 2023).

model in Equation 16, can grow exponentially in the dimension $n$ of $\mathcal{X}$. However, $N$ is in reality chosen by whoever builds the model. Rather than using all possible subsets, a better strategy is to use prior knowledge about the problem at hand to identify meaningful interactions. Examples include using adjacent or nearby pixels in images, or small sequences of observations in time series. In this way, the computational cost of Algorithm 1 can easily be controlled, and even be held constant as the dimensionality of the data increases. The experiments of Section 5.2 illustrate this.

### 3.1 Comparison to other Shapley value algorithms

Table 2 presents the computational complexity of the various Shapley value algorithms mentioned in this paper. As we see, SHAP has an exponential complexity in the dimensionality of the data (the number of variables), and all other algorithms must make concessions to escape this prohibitive computational cost. One way is to assume a specific form for the model (TreeSHAP, STAR-SHAP). The other is to give up the exactness of the values returned by the algorithm. The most salient difference between STAR-SHAP and the model-agnostic algorithms is that STAR-SHAP only needs to calculate the output of the partial models ($f_I$ in Equation (16)), entirely forgoing an explicit dependency in the dimensionality of the data. When this cost $F(k)$ is much smaller than the cost $F(n)$ of calculating the output of the entire model, for instance in large, high-dimensional ones, we should expect to see a notably faster runtime for STAR-SHAP. Combined with the exactness of the values returned by STAR-SHAP, we believe it to be the algorithm to use in the context of STAR models. We run experiments in Section 5.2 to demonstrate these advantages of STAR-SHAP over other algorithms.

### 3.2 Applications of STAR-SHAP

STAR-SHAP can calculate the Shapley values, as defined in Equation (23), for any STAR model. This includes Generalized Additive Models (GAMs) (Hastie, 2017), Generalized Additive Models with Pairwise Interactions (GA2Ms) (Lou et al., 2013), Sparse Polynomial Regression (Huang et al., 2010), Sparse Decision Trees (Lin et al., 2020), and others. GAMs and Trees already have dedicated algorithms for computing their Shapley values (see Section 2.2), but for the others, Algorithm 1 can do so as well, as long as the number of variable interactions is moderate. Finally, another family of STAR models consists of STAR RKHS Weightings, which we introduce in the next section.

|  | RWSign | RWRelu | RWStumps |
|---|---|---|---|
| $\mathcal{W}$ | $\mathbb{R}^n$ | $\mathbb{R}^n$ | $\{1, \dots, n\} \times \mathbb{R}$ |
| $\phi(w, x)$ | $\mathrm{sign}(\langle w, x \rangle)$ | $\max(0, \langle w, x \rangle)$ | $\mathrm{sign}(x_{w_1} - w_2)$ |
| $\mathcal{K}(u, w)$ | $\exp\left(-\frac{\|u-w\|_2^2}{2\gamma^2}\right)$ | $\exp\left(-\frac{\|u-w\|_2^2}{2\gamma^2}\right)$ | $\mathbb{1}[u_1 = w_1]\exp\left(-\frac{(u_2-w_2)^2}{2\gamma^2}\right)$ |
| $p$ | $\mathcal{N}(0, \sigma^2 I_n)$ | $\mathcal{N}(0, \sigma^2 I_n)$ | $\mathcal{U}(\{1, \dots, n\}) \times \mathcal{N}(0, \sigma^2)$ |

Table 3: Three RKHS Weightings instantiations.

## 4 Structured Additive RKHS Weightings of Functions

In this section, we introduce a new RKHS Weighting instantiation better suited to regression than those found in Dubé & Marchand (2024), and show how to transform any generic instantiation of a certain form into a Structured Additive Regression (STAR) model.

### 4.1 ReLU instantiation

An RKHS Weighting instantiation is a tuple $(\mathcal{W}, \phi, \mathcal{K}, p)$ of a parameter space $\mathcal{W}$, base predictor $\phi$, kernel $\mathcal{K}$ (implying an RKHS $\mathcal{H}$) and distribution $p$. For example, instantiation I1 of Dubé & Marchand (2024) (which we call RWSign) uses $\mathcal{W} = \mathcal{X} = \mathbb{R}^n$, $\phi(w, x) = \mathrm{sign}(\langle w, x \rangle)$, the gaussian kernel $\mathcal{K}(u, w) = \exp\left(-\frac{\|u-w\|_2^2}{2\gamma^2}\right)$, and a normal distribution $p = \mathcal{N}(0, \sigma^2 I_n)$.

The instantiation that we propose, which we name RWRelu, consists simply in replacing the base predictor of RWSign by the rectified linear unit (ReLU), i.e. $\phi(w, x) = \mathrm{ReLU}(\langle w, x \rangle) = \max(0, \langle w, x \rangle)$. (See Table 3 for a quick overview of each instantiation.) In order to calculate the output of the model, which takes the form:

$$\Lambda\alpha(x) = \sum_{t=1}^{T} a_t \mathop{\mathbb{E}}_{w \sim p} [\mathcal{K}(w_t, w)\phi(w, x)], \tag{24}$$

for some $\alpha = \sum_{t=1}^{T} a_t \mathcal{K}(w_t, \cdot) \in \mathcal{H}$, we need an analytical form for the expectation in Equation (24). We encapsulate that formula in the following theorem.

**Theorem 4.1.** *Consider the RKHS Weighting instantiation* $(\mathcal{W}, \phi, \mathcal{K}, p)$ *where* $\mathcal{W} = \mathcal{X} = \mathbb{R}^n$, $\phi(w, x) = \max(0, \langle w, x \rangle)$, $\mathcal{K}(u, w) = \exp\left(-\frac{\|u-w\|_2^2}{2\gamma^2}\right)$, $p = \mathcal{N}(0, \sigma^2 I_n)$. *Then we have:*

$$\mathop{\mathbb{E}}_{w \sim p} [\mathcal{K}(u, w)\phi(w, x)] =$$

$$\left(1 + \frac{\sigma^2}{\gamma^2}\right)^{-n/2} e^{\frac{-\|u\|_2^2}{2\sigma^2 + 2\gamma^2}} \frac{\|x\|}{2\sqrt{\pi}} \left(\sqrt{2}\zeta e^{-\frac{\langle u', x \rangle^2}{2\zeta^2 \|x\|^2}} + \sqrt{\pi}\frac{\langle u', x \rangle}{\|x\|}\left[1 + \mathrm{erf}\left(\frac{\langle u', x \rangle}{\sqrt{2}\zeta\|x\|}\right)\right]\right), \tag{25}$$

*where* $\zeta$ *is defined by the relationship* $\frac{1}{2\zeta^2} = \frac{1}{2\gamma^2} + \frac{1}{2\sigma^2}$ *and* $u' := \left(1 + \frac{\gamma^2}{\sigma^2}\right)^{-1} u$.

*Proof.* The calculus can be found in the appendix. $\square$

Dubé & Marchand (2024) also define some instantiation-dependent theoretical constants (see Theorem 4 of Dubé & Marchand (2024)) which are then used in the theoretical guarantees. We provide those values for RWRelu and the relevant calculus in the appendix. Importantly, this means that all theoretical guarantees proved in Dubé & Marchand (2024) also apply to RWRelu.

### 4.2 RKHS Weightings as STAR models

Instantiating an RKHS Weighting model consists in choosing its four components $(\mathcal{W}, \phi, \mathcal{K}, p)$. We will show how to instantiate an RKHS Weighting which is also a STAR model, first in full generality, and then by

transforming an already existing instantiation into a STAR model. By having a STAR model which is also an RKHS Weighting, we keep all of the benefits of the RKHS Weightings, i.e. its theoretical guarantees, learning algorithms, pruning methods (Dubé & Marchand, 2024), but also gain interpretability through the computation of the Shapley values, using Algorithm 1.

The defining characteristic of STAR models is that they are built with functions that use only a subset of the variables at a time. The key to defining an RKHS Weighting which is also a STAR model is therefore the base predictor $\phi$. By choosing a base predictor which can flexibly change the features that it considers, we can obtain a STAR model. We now go through each ingredient and see how they must be defined.

1. The parameter space $\mathcal{W}$ is accompanied by the space of feature subsets $\mathcal{P}([n])$ (the family of all subsets of $[n]$).

2. The base predictor takes as input both a parameter $w$ and a feature subset $I$, as well as an instance $x$; $\phi((w,I),x) := \phi_I(w, x_I)$ could be any predictor defined on the feature subset $I$.

3. The kernel must take into account the feature subsets, meaning it is now a function of the form $\mathcal{K} : (\mathcal{W} \times \mathcal{P}([n])) \times (\mathcal{W} \times \mathcal{P}([n])) \to \mathbb{R}$.

4. The distribution $p$ on $\mathcal{W} \times \mathcal{P}([n])$ generates both parameters and feature subsets.

All in all, the instantiation takes the form $(\mathcal{W} \times \mathcal{P}([n]), \phi, \mathcal{K}, p)$. Now, to show that this is indeed a STAR model. Given a weight function $\alpha = \sum_{t=1}^{T} a_t \mathcal{K}((w_t, I_t), \cdot)$, the output of the predictor on an arbitrary instance $x$ is given by:

$$
\begin{aligned}
\Lambda\alpha(x) &= \underset{(w,I)\sim p}{\mathbb{E}} [\alpha(w,I)\phi((w,I),x)] \\
&= \sum_{t=1}^{T} a_t \underset{(w,I)\sim p}{\mathbb{E}} [\mathcal{K}((w_t, I_t), (w,I))\phi_I(w, x_I)] \\
&= \sum_{J\subseteq[n]} \sum_{t:I_t=J} a_t \underset{(w,I)\sim p}{\mathbb{E}} [\mathcal{K}((w_t, I_t), (w,I))\phi_I(w, x_I)].
\end{aligned}
\tag{26}
$$

We have thus expressed the predictor as a sum of functions defined on feature subsets. This is therefore a STAR model, with the output on an arbitrary $x$ given by $\Lambda\alpha(x) = \sum_{J\subseteq[n]} f_J(x_J)$, with:

$$
f_J(x_J) := \sum_{t:I_t=J} a_t \underset{(w,I)\sim p}{\mathbb{E}} [\mathcal{K}((w_t, I_t), (w,I))\phi_I(w, x_I)].
\tag{27}
$$

This framework for expressing an RKHS Weighting as a STAR model is in full generality. Of course, prior knowledge specific to a given problem can be inserted into the choices for a base predictor, kernel or distribution. For instance, we might be interested only in sequences of features when presented with text, time series, or other sequential data, in which case the distribution $p$ can simply put a weight of 0 to any non-sequential feature subset. This family of STAR models is therefore highly flexible.

### 4.3 Transforming an existing instantiation into a STAR model

Perhaps the simplest way to instantiate an RKHS Weighting as a STAR model using the framework described in the previous section is to take an existing instantiation $(\mathcal{W}, \phi, \mathcal{K}_{\mathcal{W}}, p_{\mathcal{W}})$ which satisfies a few simple conditions:

1. $\mathcal{X} = \mathcal{W} = \mathbb{R}^n$.

2. The base predictor $\phi(w, x)$ is a function of $\langle w, x \rangle$, i.e. $\phi(w, x) = \phi(\langle w, x \rangle)$.

3. The distribution $p_{\mathcal{W}}$ on $\mathcal{W}$ generates each component independently.

| | Existing instantiation | STAR version |
|---|---|---|
| Parameter space | $\mathcal{W}$ | $\mathcal{W} \times \mathcal{P}([n])$ |
| Base predictor | $\phi(w,x) = \phi(\langle w, x \rangle)$ | $\phi((w,I),x) = \phi(\langle w_I, x_I \rangle)$ |
| Kernel | $\mathcal{K}_{\mathcal{W}}$ | $\mathcal{K}((w_1, I_1),(w_2, I_2)) = \mathcal{K}_{\mathcal{W}}(w_1, w_2)\mathbb{1}[I_1 = I_2]$ |
| Distribution | $p_{\mathcal{W}}$ | $p(w,I) = p_{\mathcal{W}|I}(w)p_{[n]}(I)$ |

Table 4: Cheat sheet for transforming an existing RKHS Weighting instantiation into a Structured Additive Regression model.

In that situation, modifying the base predictor to work on feature subsets is trivially easy:

$$\phi((w,I),x) := \phi(\langle w_I, x_I \rangle). \tag{28}$$

In other words, simply ignore all the variables not in the feature subset. As for the distribution $p$, the independence allows us to marginalize away the unwanted variables. The distribution $p(w, I)$ is the product $p_{\mathcal{W}|I}(w)p_{[n]}(I)$, where $p_{[n]}$ is a chosen distribution over the feature subsets, and $p_{\mathcal{W}|I}(w)$ is the original distribution $p_{\mathcal{W}}$ marginalized to the features in $I$. Finally, we elect to use the indicator kernel $\mathcal{K}_{[n]}(I_1, I_2) := \mathbb{1}[I_1 = I_2]$ over the feature subsets, and use the kernel product $\mathcal{K}((w_1, I_1),(w_2, I_2)) := \mathcal{K}_{\mathcal{W}}(w_1, w_2)\mathcal{K}_{[n]}(I_1, I_2)$.[4] We end up with the new instantiation $(\mathcal{W} \times \mathcal{P}([n]), \phi, \mathcal{K}_{\mathcal{W}} \times \mathcal{K}_{[n]}, p)$. (Table 4 summarizes all of this.) Assuming a weight function $\alpha = \sum_{t=1}^{T} a_t \mathcal{K}((w_t, I_t), \cdot)$, the model is:

$$\Lambda\alpha(x) = \sum_{J \subseteq [n]} \sum_{t: I_t = J} a_t p_{[n]}(I_t) \underset{w \sim p_{\mathcal{W}|I_t}}{\mathbb{E}} [\mathcal{K}_{\mathcal{W}}(w_t, w)\phi(\langle w, x_{I_t} \rangle)]. \tag{29}$$

The only element that still needs to be further specified is the distribution $p_{[n]}$ on the feature subsets. We list below the two most obvious candidates, but note that there is complete flexibility here for the user of the model to specify any distribution that they deem adequate for the dataset at hand.

1. The simplest option is to set $p_{[n]}$ as the uniform distribution on feature subsets of size equal to $k$, where $k$ becomes a hyperparameter of the algorithm. A slight variation of this is to use feature subsets of size up to $k$, allowing a larger variety of variable interactions.

2. Some datasets, like time series or genomic data, have structure, where adjacent features are correlated in some meaningful way. When presented with such sequential data, it is natural to consider sequences of features. We can then set $p_{[n]}$ as the uniform distribution over sequences of length $k$ (or length up to $k$).

Because of the presence of the factor $p_{[n]}(I_t)$ in Equation (29), using the first option (arbitrary subsets of size $k$) with large $k$ is not recommended. Indeed, since $p_{[n]}(I_t) = 1/\binom{n}{k} \approx \frac{1}{n^k}$, the model output tends to zero exponentially as $k$ increases. Large coefficients $a_t$ are then required to offset the small density. This leads to a large norm $\|\alpha\|_{\mathcal{H}}$ of the weight function, and therefore poorer guarantees (Dubé & Marchand, 2024). More intuitively, it is important that a significant portion of the feature subsets $I_t$ in Equation (29) are meaningful interactions. Otherwise, the partial predictors built on these uninformative interactions will themselves be poor. Reducing the size of the space of feature subsets by limiting the distribution $p_{[n]}$ to mostly meaningful interactions is crucial to get a well-performing STAR RKHS Weighting model. Sequential feature subsets achieve this. Since there are only $n - k + 1$ sequences of features of length $k$ in an $n$-dimensional dataset, $p_{[n]}$ will always be of the order of $\frac{1}{n}$. When considering sequences of length up to $k$, then $p_{[n]}$ will be about $\frac{1}{kn}$, which is undeniably preferable over $\frac{1}{n^k}$.

---

[4] We use the indicator kernel for two related but distinct reasons. The first is the assumption that, in general, the comparison between parameters, i.e. the kernel evaluation $\mathcal{K}_{\mathcal{W}}(w_1, w_2)$, only makes sense if $w_1$ and $w_2$ are parameters over the same features. Second, it ensures that $w_1$ and $w_2$ are the same dimensionality, so that the kernel evaluation $\mathcal{K}_{\mathcal{W}}(w_1, w_2)$ is well-defined. There is some abuse of notation here, as we use the same symbol $\mathcal{K}_{\mathcal{W}}$ to denote a function over potentially different spaces (e.g. $w_1$ could be from $\mathbb{R}^3$, and $w_2$ from $\mathbb{R}^4$), but deem it benign enough, as it does simplify the expressions.

We can use this transformation to immediately obtain STAR models based on instantiations RWSign and RWRelu, which can be learned using any algorithm from Dubé & Marchand (2024). (Note that instantiation RWStumps is already a GAM, so no transformation is required). This makes up a new family of STAR models, which are easy to train, as they do not require explicitly specifying feature interactions when using one of the learning algorithms from Dubé & Marchand (2024).

(For completeness, we describe in the appendix how to obtain the theoretical constants from Dubé & Marchand (2024) for a STAR RKHS model. Once again, this means that the theoretical results of Dubé & Marchand (2024) apply to the STAR RKHS Weightings constructed in this way.)

## 5 Experiments

We introduced three new tools in this paper: STAR-SHAP, an algorithm for calculating the Shapley values of a STAR model; RWRelu, a new RKHS Weighting instantiation; and a new family of STAR models, obtained by transforming existing RKHS Weighting instantiations into STAR models. We present in the next few sections three experiments to test these new tools. First, we compare the computation time of STAR-SHAP to other Shapley value algorithms in Section 5.2. Then, Section 5.3 presents a simple regression performance experiment, designed to compare various regression algorithms to the new RWRelu and STAR RKHS Weighting models. Next, the experiment of Section 5.4 investigates a scenario (time series regression) in which we expect the STAR RKHS Weightings to be better suited than the simple regression problems of the Section 5.3 experiment. Finally, Section 5.5 compares the explicit Shapley values of a STAR RKHS model to those of a conventional model, both trained on the same dataset.

### 5.1 Generating synthetic datasets

To generate our synthetic datasets, we took inspiration from the sum of unanimity models used by Fumagalli et al. (2023). For given values of $k$ (the size of the feature subsets), $n$ (the total number of variables), and $T$ (the desired number terms), we generate a $k$-STAR synthetic model over boolean instances ($\mathcal{X} = \{\pm 1\}^n$) as $f(x) = \sum_{i=1}^{T} a_i \mathbb{1}[\forall j \in I_i, x_j = 1]$, where $I_1, \ldots, I_T$ are uniformly and independently generated feature subsets of size $k$, and $a_1, \ldots, a_T$ are coefficients uniformly generated in $[0, 1]$. To go along with every random model, we also generate a random dataset of boolean-valued instances $\mathcal{S} \in \mathcal{X}^m$, where each component of an instance $x$ is independently and uniformly sampled from $\{\pm 1\}$.

### 5.2 Computation time of the Shapley values

Figure 1 shows the computation time of the Shapley values of 5-STAR models (that is, with five-way interactions between variables) using various algorithms. We notice, of course, the exponential growth of the computation time for SHAP, while STAR-SHAP cares little for the dimensionality when the size of the feature subsets is fixed. For completeness, we also show the computation time growth of using STAR-SHAP to calculate the Shapley values of a model which uses the full $n$-way interactions. We observe the expected exponential behavior, mirroring SHAP. Meanwhile, Figure 1b clearly illustrates the fact that STAR-SHAP lacks an explicit dependency on the dimensionality of the data, making it much more efficient than other algorithms at higher dimensionalities.

However, Figure 1 does not account for the quality of the approximations, and only looks at 5-STAR models. Unbiased KernelSHAP and SHAP-IQ are not so much slower than STAR-SHAP, but how good are their approximations of the exact Shapley values? Since Unbiased KernelSHAP and SHAP-IQ coincide when calculating 1-Shapley values (Fumagalli et al., 2023), possess theoretical guarantees, and have better empirical performance than KernelSHAP (according to Fumagalli et al. (2023)), we assess the quality of SHAP-IQ in Figure 2. The figure shows how the advantage of STAR-SHAP increases with dimensionality. Indeed, not only does the computation time of SHAP-IQ increase, but also the quality of its approximations degrades, requiring additional computation time to return good approximations. Meanwhile, the values returned by STAR-SHAP are always exact, and are calculated efficiently.

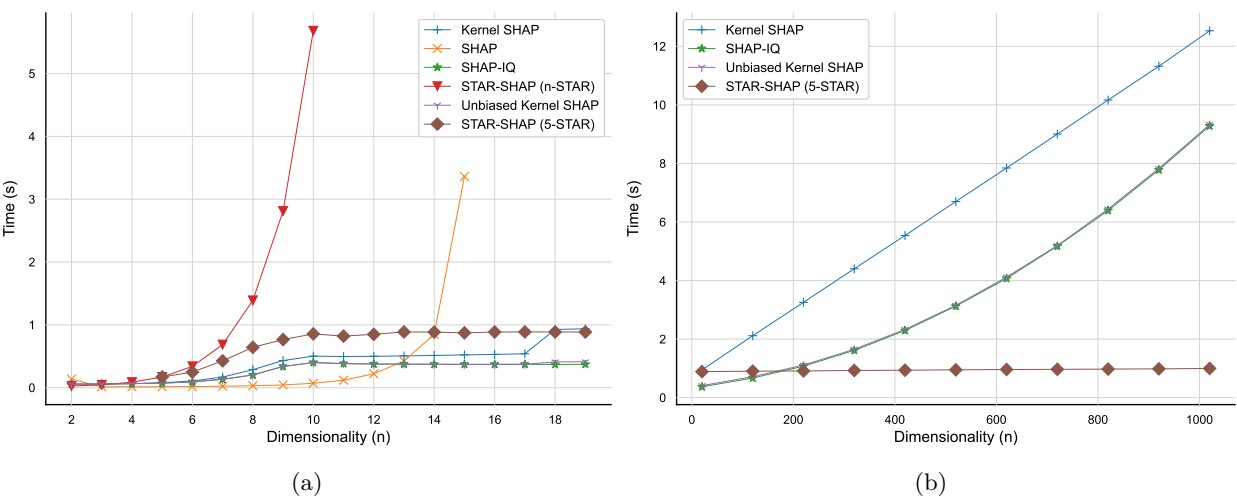

Figure 1: Computation time of the Shapley values of a Structured Additive Regression model with respect to the dimensionality of the dataset. For every value of $n$, a 5-STAR random synthetic model with $T = 50$ terms was generated as described in Section 5.1, along with a dataset of 10 examples. Then, all algorithms were used to calculate the Shapley values of the model for all variables and examples. Similarly, $n$-STAR models, using all $n$ possible interactions, were generated, and STAR-SHAP applied to this model. SHAP and the approximation algorithms were also applied to this model for values $n < 5$, for which a 5-STAR model cannot exist. Figure 1b is the rescaled continuation of Figure 1a without the exponentially increasing algorithms. Every point is the average of 20 runs to smooth out any possible variability due to hardware or CPU conflicts. We used the SHAP-IQ implementation (Muschalik et al., 2024) of KernelSHAP, Unbiased KernelSHAP and SHAP-IQ itself, with a budget of 256. Note that the complexity of KernelSHAP appears linear at this scale, but it is in fact cubic in $n$ (see Table 2).

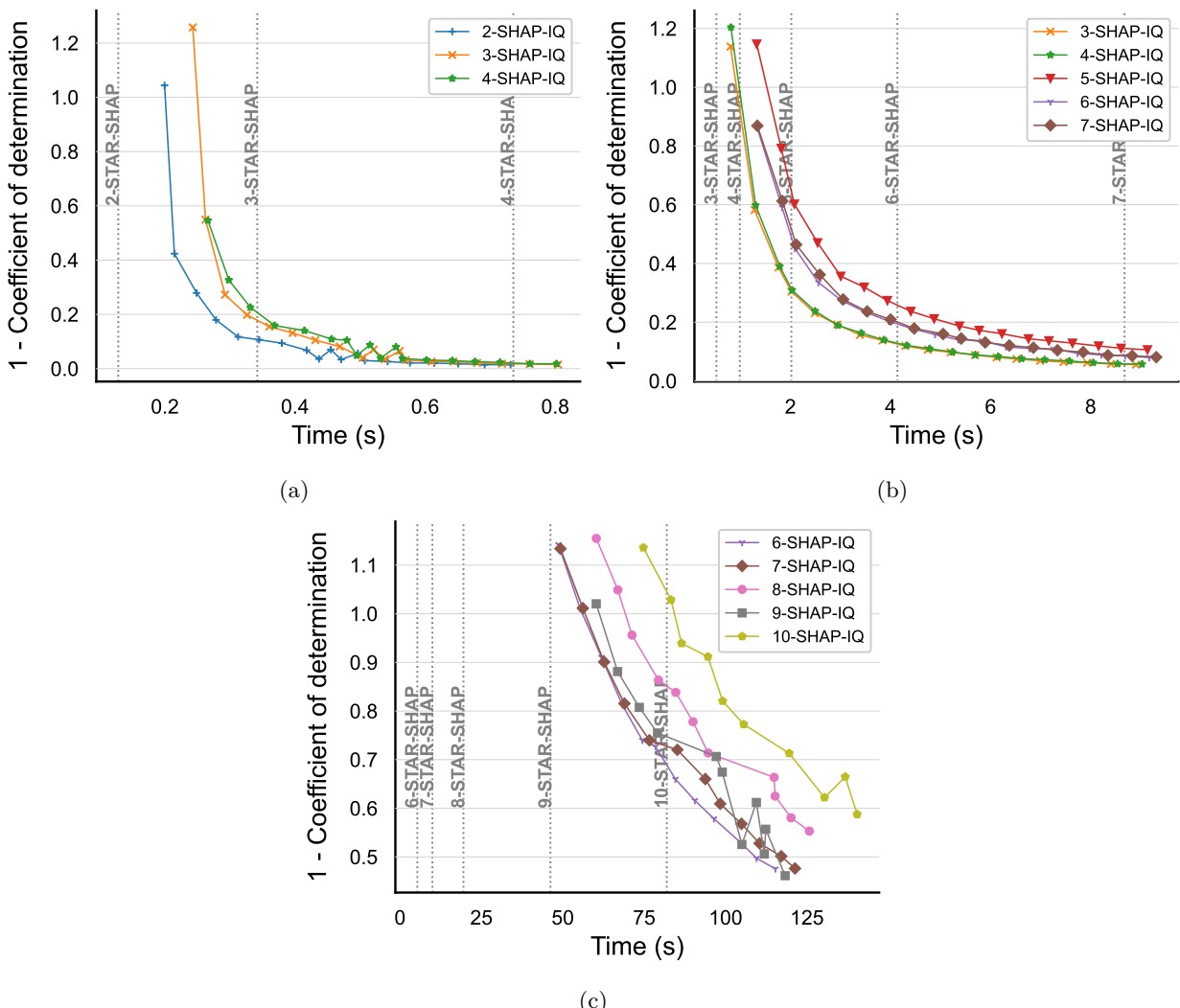

Figure 2: One minus the coefficient of determination ($R^2$) of the Shapley values of synthetic models approximated by SHAP-IQ with regard to the computation time (specified indirectly through the budget hyperparameter). The models were generated as described in Section 5.1, with a fixed number of terms in the models ($T = 100$), and varying dimensionality $n$. For each value of $k$, the size of the feature subsets in the STAR model, the Shapley values of a synthetic model on a dataset of size 10 were calculated approximately by SHAP-IQ ($k$-SHAP-IQ), and exactly by STAR-SHAP ($k$-STAR-SHAP). The $R^2$ was then calculated between the vectorized $10 \times n$ matrices of approximated and exact Shapley values. A vertical dotted line labeled $k$-STAR-SHAP indicates the time STAR-SHAP took to calculate the exact Shapley values for that value of $k$. Every point is the average of 10 independently seeded runs. Figure 2a: on low-dimensionality datasets ($n = 10$ variables, $T = 100$ terms in the model), SHAP-IQ starts generating Shapley values with an $R^2$ coefficient larger than 0 faster than STAR-SHAP at $k = 3$. This means that STAR-SHAP is the better algorithm choice for $k \leq 3$, and then gradually losses its merits as $k$ increases and SHAP-IQ is allowed more computation time. In Figure 2b, more total variables were used in the models ($n = 100$, $T = 100$). Now, STAR-SHAP is unambiguously preferable up to about $k = 5$, after which SHAP-IQ starts becoming competitive if perfectly exact Shapley values are not required. Finally, Figure 2c used $n = 500$ total variables (with $T = 100$ terms) to build the models. Even for $k = 10$, which is an unusually high number of variable interactions for a model, SHAP-IQ generates barely acceptable Shapley values when given the same amount of time as STAR-SHAP. For any smaller value of $k$ and the same computation time as STAR-SHAP, SHAP-IQ yielded extremely poor quality approximations of the Shapley values.

|              | Training size | Test size | Dimensionality | Source (clickable) |
|--------------|---------------|-----------|----------------|--------------------|
| abalone      | 3132          | 1045      | 10             | UCI                |
| diabetes     | 331           | 111       | 10             | Scikit-learn       |
| housing      | 15480         | 5160      | 8              | Scikit-learn       |
| concrete     | 772           | 258       | 8              | UCI                |
| conductivity | 15947         | 5316      | 81             | UCI                |
| wine         | 133           | 45        | 13             | UCI                |

Table 5: Datasets of the regression experiment.

Of course, these computation times and accuracies are highly dependent on the models and datasets that were generated, but they strongly hint at a general trend. If the size of the feature subsets is limited, then the computation time of STAR-SHAP will be fast. Since it always returns exact values, it will be the preferable algorithm in that scenario. This is true regardless of model size. However, the more total variables in the model, the more feature interactions can be present before needing to consider using an approximation algorithm, due to their slower convergence rate at higher dimensionalities. This makes STAR-SHAP especially valuable in the presence of high-dimensional datasets.

## 5.3 Regression performance

Three new models arise from Section 4: the RWRelu, and the STAR versions of RWSign and RWRelu. This calls for a simple regression experiment comparing the performance of these models to more bread and butter regression techniques, especially interpretable ones. Using 5-fold cross-validation to select hyperparameters, we trained a variety of models on several regression datasets taken from the UCI Machine Learning Repository (Dua & Graff, 2017) and Scikit-learn (Pedregosa et al., 2011). See Table 5 for information on the datasets, and the appendix for the detailed experimental setup.

We can look to the results in Table 6 to answer a few important questions.

**Q1.** How does RWRelu, the ReLU version of an RKHS Weighting, compare to RWSign and RWStumps, the two original instantiations, for regression tasks?

**R1.** We can see that RWRelu performs equally or better than RWSign on all but one dataset (Wine). On the other hand, RWStumps outperforms RWRelu on 4 of the 6 datasets. The STAR versions of RWRelu and RWSign outperform the other (for equal $k$) on some datasets, and not others, neither systematically standing out. These observations give some weight, but not much, to the conjecture that RWRelu is better suited to regression than the other two RKHS Weighting instantiations. Rather, they might all be complementary, each with better and worse use cases.

**Q2.** How do the STAR versions of RKHS Weighting models perform in comparison to their original counterparts?

**R2.** There is a moderate loss of performance going from RWRelu to its 1-STAR version, a cost that might be acceptable to pay for the immense gain in interpretability. However, increasing $k$ does improve the performance significantly on all but one dataset, Conductivity, and 3-STAR RWRelu is the best performing STAR version of the model, which shows that adding interactions can improve the quality of the model. The extremely poor performance of 3, 4 and 5-STAR models on Conductivity is caused by the higher dimensionality of the dataset. (See this footnote[5] for the in-depth explanation.) Indeed, larger feature subsets should not blindly be used except on low-dimensional datasets, or failure of learning might occur. As for the STAR versions of RWSign, they are less convincing than RWRelu, with the 1-STAR version in particular exhibiting extremely poor performance overall.

---

[5]We recall the presence of a factor $p_{[n]}(I_t)$ in Equation (29), the expression for a STAR RKHS Weighting. When using the uniform distribution over all possible feature subsets of size $k$, we have $p_{[n]}(I_t) = 1/\binom{n}{k} \approx \frac{1}{n^k}$, which is an extremely small value for high-dimensionality datasets, leading to vanishing outputs of the model. And indeed, the offending dataset, Conductivity, is far higher-dimensional than the other ones, proving the point.

Table 6: Comparison of the regression performance of STAR RKHS Weightings and the ReLU instantiation, learned using Algorithm 3 of Dubé & Marchand (2024) (the Least Squares fit) with $T = 5000$, to a variety of other regression algorithms. $k$-STAR signifies the STAR version of the model using feature subsets of size $k$. The table values are the test $R^2$ (coefficient of determination). The highest (best) values for each dataset have been bolded.

| Dataset
Algorithm | abalone | diabetes | housing | concrete | conductivity | wine |
|---|---|---|---|---|---|---|
| DecisionTreeRegressor | 0.499 | 0.105 | 0.677 | 0.796 | 0.882 | 0.867 |
| EBM | 0.552 | 0.331 | **0.822** | **0.931** | **0.903** | 0.897 |
| KernelRidge | **0.587** | 0.282 | 0.771 | 0.831 | 0.872 | **0.951** |
| LinearRegression | 0.543 | 0.359 | 0.591 | 0.623 | 0.736 | 0.804 |
| RWRelu | 0.577 | 0.349 | 0.734 | 0.835 | 0.842 | 0.832 |
| RWRelu 1-STAR | 0.555 | 0.314 | 0.662 | 0.780 | 0.777 | 0.808 |
| RWRelu 2-STAR | 0.577 | 0.305 | 0.731 | 0.847 | 0.805 | 0.843 |
| RWRelu 3-STAR | 0.580 | 0.332 | 0.744 | 0.847 | 0.568 | 0.845 |
| RWRelu 4-STAR | 0.579 | 0.350 | 0.737 | 0.846 | 0.032 | 0.835 |
| RWRelu 5-STAR | 0.578 | 0.334 | 0.729 | 0.844 | 0.001 | 0.837 |
| RWRelu [1, 2, 3, 4, 5]-STAR | 0.577 | 0.298 | 0.732 | 0.808 | 0.779 | 0.826 |
| RWSign | 0.578 | 0.348 | 0.711 | 0.828 | 0.728 | 0.893 |
| RWSign 1-STAR | 0.350 | 0.302 | 0.354 | 0.471 | 0.685 | 0.750 |
| RWSign 2-STAR | 0.579 | 0.369 | 0.696 | 0.835 | 0.816 | 0.868 |
| RWSign 3-STAR | **0.586** | 0.339 | 0.720 | 0.840 | 0.468 | 0.912 |
| RWSign 4-STAR | 0.585 | 0.349 | 0.708 | 0.794 | 0.003 | 0.896 |
| RWSign 5-STAR | 0.580 | 0.333 | 0.705 | 0.770 | -0.000 | 0.899 |
| RWSign [1, 2, 3, 4, 5]-STAR | 0.580 | **0.399** | 0.698 | 0.826 | 0.694 | 0.840 |
| RWStumps | 0.567 | 0.339 | 0.768 | 0.911 | 0.859 | 0.864 |
| SVR | 0.564 | 0.293 | 0.770 | 0.799 | 0.862 | **0.950** |

|  | Training size | Test size | Dimensionality |
|---|---|---|---|
| ChlorineConcentration | 6990 | 57585 | 10 |
| Computers | 16434 | 16434 | 10 |
| ECG5000 | 6487 | 58487 | 10 |
| FacesUCR | 2388 | 24588 | 10 |
| LargeKitchenAppliances | 24684 | 24684 | 10 |
| MelbournePedestrian | 3579 | 7314 | 10 |

Table 7: Datasets of the time series prediction experiment.

**Q3.** How does the new RWRelu instantiation perform compared to other regression algorithms?

**R3.** The two longstanding, bread and butter regression algorithms that are Kernel Ridge Regression and Support Vector Regression outperform RWRelu respectively on 4 of the 6, and 3 of the 6 datasets. This is far from an indictment of RWRelu.

**Q4.** Can interpretable RKHS Weightings perform similarly or better than other interpretable methods?

**R4.** The Explainable Boosting Machine (EBM) of (Lou et al., 2013) (software package described in Nori et al. (2019)) appears overall to be the best at both accuracy and interpretability, at least in this experiment. Kernel Ridge Regression is best on two datasets, and SVR, 3-STAR RWSign, and 1-to-5 STAR RWSign are each best on one dataset. Therefore, we can say that interpretable RKHS Weightings are at least sometimes competitive.

To conclude this experiment, we see that transforming an RKHS Weighting into a STAR model is a valid way of learning an interpretable model, at a relatively small cost of performance. However, other methods, especially Explainable Boosting Machines, often outperform these RKHS Weightings in both accuracy and interpretability.

## 5.4 Time series prediction performance

Perhaps the biggest takeaway from Table 6 is the excellence of the Explainable Boosting Machine on both the accuracy and interpretability fronts. The RKHS Weightings did not shine particularly brightly. However, the most important characteristic of RKHS Weightings is their high flexibility in instantiating the model. We can improve performance by utilizing an instantiation better adapted to the problem at hand. In the specific case of STAR RKHS Weightings, a tool we can play with is the distribution on the feature subsets. For instance, instead of using the uniform distribution over all subsets of a certain length, we can use a uniform distribution over *sequences* of adjacent features (e.g. features $(2, 3, 4)$, or $(n - 2, n - 1, n)$). This makes the STAR RKHS instantiation much better suited to solve regression problems for which adjacent data features are correlated, such as time series. Therefore, a better comparison between STAR RKHS Weightings and the Explainable Boosting Machine is on such datasets, using a distribution over sequences of features. This is what we do here.

Specifically, we run both the Explainable Boosting Machine (EBM) and STAR RKHS Weightings on time series datasets taken from the UCR Archive (Dau et al., 2018). Each dataset was cut into segments of length 11, so that each instance $x$ has length 10, and the value to predict $y$ is the 11-th value. See Table 7. Hyperparameters were selected by 5-fold cross-validation. See the appendix for the detailed experimental setup.

The results of this experiment can be found in Table 8. The RKHS Weightings fare much better in this experiment than the previous. Here, they (slightly) outperform the EBM on 5 of the 6 datasets. On the other hand, the EBM does systematically reach the highest training $R^2$, suggesting that some overfitting has taken place, even though the parameters were chosen by cross-validation. Perhaps a different set of candidate parameters would have led to a smaller generalization gap. Finally, we notice the expected trend that larger feature subsets on the STAR RKHS Weightings lead to better results, with the full RWRelu instantiation

Table 8: Comparison of the time series prediction performance of STAR RKHS Weightings and the ReLU instantiation, learned using Algorithm 3 of Dubé & Marchand (2024) (the Least Squares fit) with $T = 5000$, to the Explainable Boosting Machine (EBM). $k$-STAR signifies the STAR version of the model using feature subsets of size $k$. The highest (best) values for each dataset have been bolded.

| Dataset | Algorithm | Training $R^2$ | Test $R^2$ | Training time (s) |
|---|---|---|---|---|
| ChlorineConcentration | EBM | **0.955** | 0.916 | 63.853 |
| | RWRelu | 0.934 | **0.928** | 3.307 |
| | RWRelu 1-STAR | 0.638 | 0.642 | 36.375 |
| | RWRelu 2-STAR | 0.813 | 0.805 | 40.390 |
| | RWRelu 3-STAR | 0.907 | 0.899 | 44.500 |
| | RWRelu 4-STAR | 0.928 | 0.920 | 48.720 |
| | RWRelu 5-STAR | 0.939 | **0.927** | 54.159 |
| | RWRelu [1, 2, 3, 4, 5]-STAR | 0.924 | 0.906 | 19.823 |
| Computers | EBM | **0.882** | 0.842 | 16.291 |
| | RWRelu | 0.821 | **0.851** | 6.085 |
| | RWRelu 1-STAR | 0.786 | 0.838 | 44.742 |
| | RWRelu 2-STAR | 0.802 | 0.840 | 50.501 |
| | RWRelu 3-STAR | 0.811 | 0.848 | 55.049 |
| | RWRelu 4-STAR | 0.814 | 0.847 | 58.744 |
| | RWRelu 5-STAR | 0.818 | 0.848 | 64.858 |
| | RWRelu [1, 2, 3, 4, 5]-STAR | 0.817 | **0.849** | 29.086 |
| ECG5000 | EBM | **0.943** | **0.886** | 23.525 |
| | RWRelu | 0.900 | 0.882 | 3.144 |
| | RWRelu 1-STAR | 0.843 | 0.841 | 36.361 |
| | RWRelu 2-STAR | 0.877 | 0.871 | 40.024 |
| | RWRelu 3-STAR | 0.883 | 0.875 | 43.769 |
| | RWRelu 4-STAR | 0.893 | 0.879 | 48.122 |
| | RWRelu 5-STAR | 0.899 | 0.882 | 53.384 |
| | RWRelu [1, 2, 3, 4, 5]-STAR | 0.894 | 0.882 | 19.157 |
| FacesUCR | EBM | **0.868** | 0.661 | 7.674 |
| | RWRelu | 0.715 | **0.704** | 1.929 |
| | RWRelu 1-STAR | 0.679 | 0.698 | 33.011 |
| | RWRelu 2-STAR | 0.690 | 0.702 | 35.613 |
| | RWRelu 3-STAR | 0.699 | **0.705** | 38.827 |
| | RWRelu 4-STAR | 0.700 | **0.705** | 42.990 |
| | RWRelu 5-STAR | 0.709 | **0.706** | 49.571 |
| | RWRelu [1, 2, 3, 4, 5]-STAR | 0.704 | **0.706** | 15.570 |
| LargeKitchenAppliances | EBM | **0.840** | 0.743 | 19.103 |
| | RWRelu | 0.813 | 0.743 | 8.775 |
| | RWRelu 1-STAR | 0.764 | 0.724 | 52.104 |
| | RWRelu 2-STAR | 0.786 | **0.748** | 56.938 |
| | RWRelu 3-STAR | 0.777 | 0.736 | 60.735 |
| | RWRelu 4-STAR | 0.803 | 0.746 | 68.894 |
| | RWRelu 5-STAR | 0.810 | 0.745 | 75.208 |
| | RWRelu [1, 2, 3, 4, 5]-STAR | 0.812 | **0.749** | 37.741 |
| MelbournePedestrian | EBM | **0.979** | **0.924** | 9.731 |
| | RWRelu | 0.946 | **0.925** | 2.335 |
| | RWRelu 1-STAR | 0.888 | 0.888 | 35.930 |
| | RWRelu 2-STAR | 0.921 | 0.902 | 36.519 |
| | RWRelu 3-STAR | 0.943 | **0.923** | 40.523 |
| | RWRelu 4-STAR | 0.943 | **0.925** | 44.806 |
| | RWRelu 5-STAR | 0.943 | **0.924** | 50.667 |
| | RWRelu [1, 2, 3, 4, 5]-STAR | 0.941 | 0.919 | 16.232 |

often being the best (although by a very small margin). Indeed, more complex interactions allow for greater expressivity of the model, so that it can better fit the data.

It is important to note that RWRelu is just one RKHS Weighting instantiation of a theoretically limitless number. It is certain that other, yet to be defined, instantiations would perform much better than RWRelu in this experiment. Indeed, it is still an open question how to craft instantiations that can perform particularly well on a given problem type (such as time series regression), or even for a specific dataset. More research is certainly required.

### 5.5 Shapley values comparison

Our final experiment shows how a STAR RKHS Weighting model could be used in practice, in comparison to a more traditional model, and how its predictions could be interpreted using Algorithm 1 (STAR-SHAP). Figure 3 compares the Shapley values of the Explainable Boosting Machine (EBM) model and the 3-STAR RWRelu model that were trained on the California Housing dataset in Section 5.3. We see that both models agree almost exactly on the order of importance of the variables, and that the bee swarms for each variable are similarly shaped between both models. This is despite the models being fundamentally different predictors: the EBM is a sum of Trees built on single variables, while the 3-STAR RKHS Weighting is a sum of partial predictors of the form given by Equation 25, using three variables each.

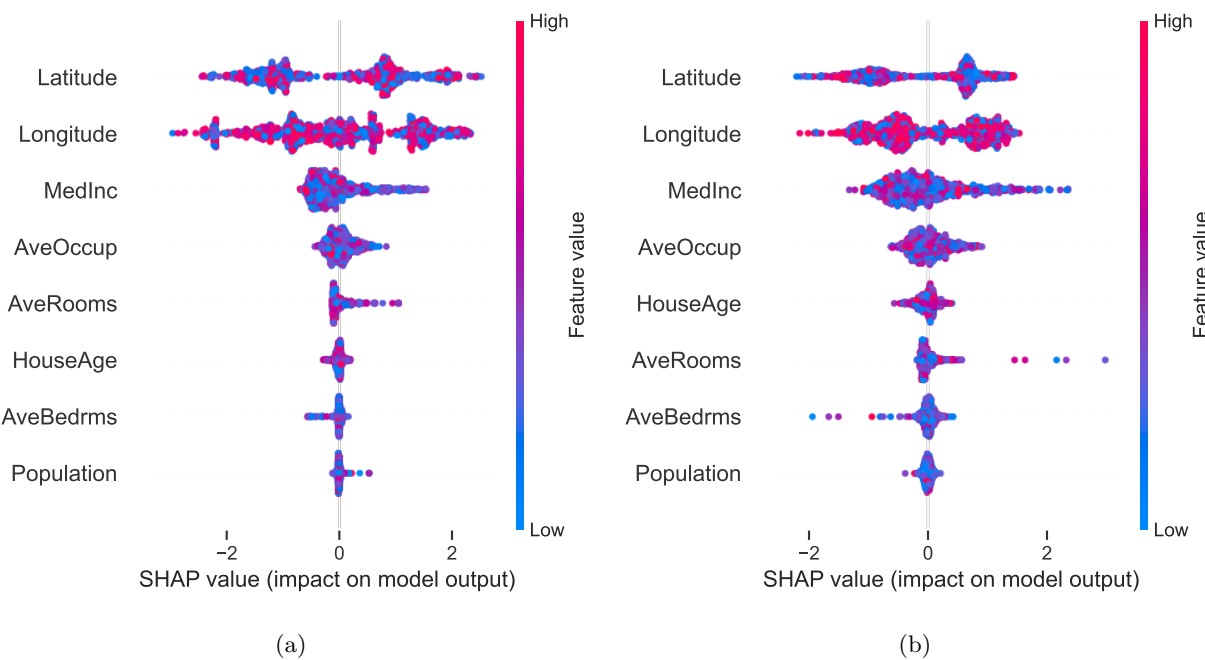

Figure 3: Beeswarm plots of the Shapley values on California Housing of the Explainable Boosting Machine (Figure 3a) and 3-STAR RWRelu (Figure 3b). Models are taken from the experiment of Section 5.3. The Shapley values of 1000 test examples were calculated using SHAP (Figure 3a) and STAR-SHAP (Figure 3b), using a background of 1000 training examples. The bee swarms, top to bottom, are in decreasing order of variable importance. This order is almost the same for both models, with only HouseAge and AveRooms being interchanged.

Figure 4 compares the Shapley values of both models on a single instance. While the predicted values are noticeably different, both models agree on the variables to consider: Longitude has a strong negative contribution to the prediction, Latitude an almost equally strong positive contribution, and AveOccup a small negative contribution. Other variables are mostly negligible. One could therefore reach the same interpretative conclusions from seeing the RKHS Weighting plot as the EBM one. This gives further credence to RKHS Weightings as a promising model family, especially an interpretable one when cast as STAR models.

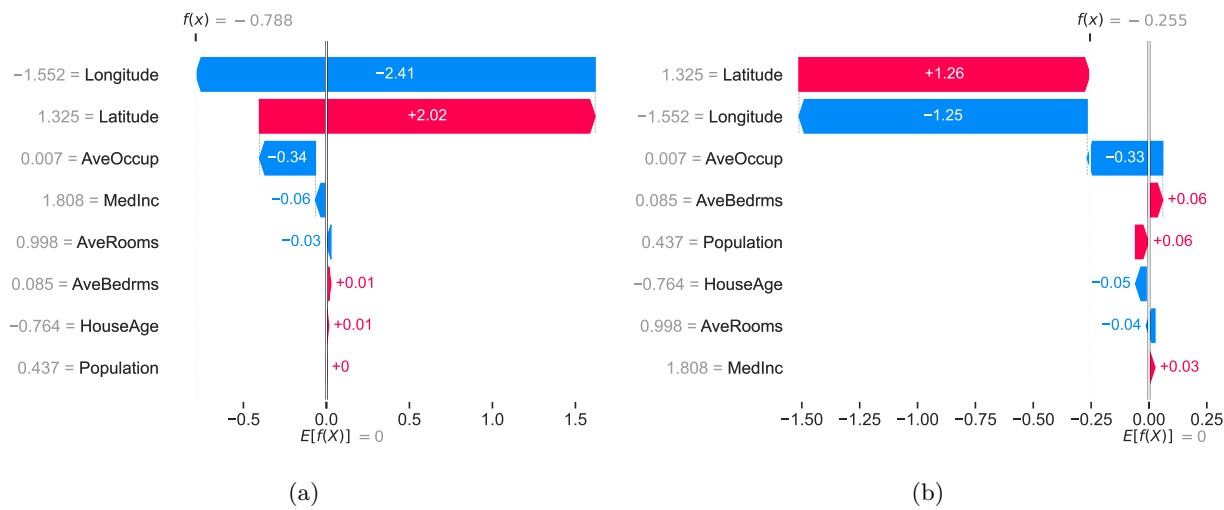

Figure 4: Waterfall plots of the Shapley values of a single instance $x$ from Figure 3. Figure 4a: Explainable Boosting Machine. Figure 4b: 3-STAR RWRelu. The true $f(x)$ value for that instance is $-0.6106$.

## 5.6 Discussion

A number of conclusions can be reached from our experiments.

Figure 1 demonstrates that STAR-SHAP is the preferred algorithm for calculating the Shapley values of a model with moderate variable interactions, especially so the more total variables are present in the data. Figure 2 more clearly shows how the advantages of STAR-SHAP increase with the total number of variables in the model, as the quality of the approximations returned by approximation algorithms degrades with dimensionality.

Table 6 shows that RKHS Weightings and STAR RKHS Weightings can reach moderately good performance on arbitrary datasets. They have a better showing in Table 8, where prior knowledge about time series led to improved performance, though it remains questionable whether RKHS Weightings should be used over conventional models.

Figures 3 and 4 show that STAR-SHAP and STAR RKHS Weightings can yield interpretable models.

All in all, while these experiments do not show that RKHS Weightings should be used over other, more conventional, tried and tested models, they are a step forward in understanding how to use RKHS Weightings, and a proof that they be high accuracy, theoretically grounded (as per Dubé & Marchand (2024)), and even interpretable models. Further research to expand the usability of RKHS Weightings, as well as find their unique strengths and applications, is warranted.

## 6 Limitations and future work

Our work presents a number of limitations, some of which can be addressed with future research.

**Algorithm 1 complexity.** In theory, Algorithm 1 can calculate the Shapley values of any STAR model, but the computational cost is exponential in the size of the feature subsets. This limits the size of the feature subsets that can be used in the model, but Figures 1 and 2 show that, for moderate interaction sizes, it is the preferred algorithm to use to calculate the Shapley values of STAR models, especially so the more total variables are in the data.

**Instantiating RKHS Weightings.** The main weakness of RKHS Weightings remains the requirement of solving a complex integral for any new instantiation. We have introduced one new instantiation, but it

is still an open problem to figure out how to access the potential of this family of models. Indeed, it is almost certain that better results than those found in Tables 6 and 8 can be obtained by using different instantiations. As Dubé & Marchand (2024) suggested, Monte Carlo approximation of the expectations might provide a good enough solution, and warrants an in-depth look.

**Prior knowledge.** As seen in Table 6, the most generic version of STAR RKHS Weightings, i.e. using the uniform distribution on all possible feature subsets of size $k$, have somewhat limited use, since their output scales as $1/\binom{n}{k}$ ($n$ being the dimensionality of the dataset). As we have already addressed, the best option is to insert prior knowledge into the model, as we have done with time series datasets by considering sequences of variables. Extending this principle to other types of structured data would increase the applicability of STAR RKHS Weighting models.

## 7  Conclusion

In this paper, we derived an efficient algorithm, named STAR-SHAP, for calculating the Shapley values of any Structured Additive Regression (STAR) model. We demonstrated its efficiency compared to other Shapley value algorithms when the number of variable interactions is limited. This advantage is further increased on high-dimensional datasets. We introduced a new RKHS Weightings instantiation, and showed how to obtain RKHS Weightings that are STAR models, giving rise to a new family of STAR models, all of which are now interpretable thanks to STAR-SHAP. We tested the prediction performance of the introduced models. While the STAR RKHS Weightings did not rise to the level of state of the art interpretable algorithms on generic regression datasets, they proved quite capable in the context of time series prediction when infused with adequate prior knowledge. Further work to increase the breadth of usable RKHS Weightings instantiations is warranted, as well as work to expand the types of datasets where RKHS Weightings can perform well.

### Acknowledgments

This work is supported by the DEEL Project CRDPJ 537462-18 funded by the Natural Sciences and Engineering Research Council of Canada (NSERC) and the Consortium for Research and Innovation in Aerospace in Québec (CRIAQ), together with its industrial partners Thales Canada inc, Bell Textron Canada Limited, CAE inc and Bombardier inc.[6]

Special thanks to the anonymous TMLR reviewers whose comments and suggestions led to significant improvements of the manuscript.

---

[6]`https://deel.quebec`

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

## A  Proofs

First, we quickly prove Equation (15).

*Proof of Equation* (15). We have:

$$\phi_i^{\text{SHAP}}(h, x) := \underset{z \sim U(\mathcal{S})}{\mathbb{E}} \, \underset{\pi \sim U(\Omega)}{\mathbb{E}} \left[ h(r_{\pi_{:i} \cup \{i\}}(z, x)) - h(r_{\pi_{:i}}(z, x)) \right]$$

$$= \underset{z \sim U(\mathcal{S})}{\mathbb{E}} \, \underset{\pi \sim U(\Omega)}{\mathbb{E}} \left[ \left( \sum_{j \in \pi_{:i} \cup \{i\}} f_j(x_j) + \sum_{j \notin \pi_{:i} \cup \{i\}} f_j(z_j) \right) - \left( \sum_{j \in \pi_{:i}} f_j(x_j) - \sum_{j \notin \pi_{:i}} f_j(z_j) \right) \right]$$

$$= \underset{z \sim U(\mathcal{S})}{\mathbb{E}} \, \underset{\pi \sim U(\Omega)}{\mathbb{E}} \left[ \left( \sum_{j \in \pi_{:i} \cup \{i\}} f_j(x_j) - \sum_{j \in \pi_{:i}} f_j(x_j) \right) + \left( \sum_{j \notin \pi_{:i} \cup \{i\}} f_j(z_j) - \sum_{j \notin \pi_{:i}} f_j(z_j) \right) \right]$$

$$= \underset{z \sim U(\mathcal{S})}{\mathbb{E}} \, \underset{\pi \sim U(\Omega)}{\mathbb{E}} \left[ \left( f_i(x_i) \right) - \left( \sum_{j \in \pi_{:i} \cup \{i\}} f_j(z_j) - \sum_{j \in \pi_{:i} \cup \{i\}} f_j(z_j) + f_i(z_i) \right) \right]$$

$$= f_i(x_i) - \underset{z \sim U(\mathcal{S})}{\mathbb{E}} \, \underset{\pi \sim U(\Omega)}{\mathbb{E}} \left[ f_i(z_i) \right]$$

$$= f_i(x_i) - \underset{z \sim U(\mathcal{S})}{\mathbb{E}} \left[ f_i(z_i) \right].$$

□

The main result to prove is Theorem 3.1, which follows from two lemmas further below.

*Proof of Theorem 3.1.* We are calculating the Shapley values of a STAR model $h(x) = \sum_{I \subseteq [n]} f_I(x_I)$, which can be written as:

$$\phi_i^{\text{SHAP}}(h, x) := \underset{z \sim U(\mathcal{S})}{\mathbb{E}} \, \underset{\pi \sim U(\Omega)}{\mathbb{E}} \left[ h(r_{\pi_{:i} \cup \{i\}}(z, x)) - h(r_{\pi_{:i}}(z, x)) \right]$$

$$= \underset{z \sim U(\mathcal{S})}{\mathbb{E}} \, \underset{\pi \sim U(\Omega)}{\mathbb{E}} \left[ \sum_{I \subseteq [n]} f_I(r_{\pi_{:i} \cup \{i\}}(z, x)_I) - \sum_{I \subseteq [n]} f_I(r_{\pi_{:i}}(z, x)_I) \right]$$

$$= \underset{z \sim U(\mathcal{S})}{\mathbb{E}} \left[ \sum_{I \subseteq [n]} \underset{\pi \sim U(\Omega)}{\mathbb{E}} \left[ f_I(r_{\pi_{:i} \cup \{i\}}(z, x)_I) - f_I(r_{\pi_{:i}}(z, x)_I) \right] \right].$$

We can immediately cut out most of the computations by noticing two facts:

1. Most functions $f_I$ are 0.

2. If $i \notin I$, then $r_{\pi_{:i} \cup \{i\}}(z, x)_I = r_{\pi_{:i}}(z, x)_I$. Hence, the entire expectation:

$$E_{\pi \sim U(\Omega)} \left[ f_I(r_{\pi_{:i} \cup \{i\}}(z, x)_I) - f_I(r_{\pi_{:i}}(z, x)_I) \right] \tag{30}$$

is 0.

The Shapley values therefore boil down to the following formula:

$$\phi_i^{\text{SHAP}}(h, x) = \underset{z \sim U(\mathcal{S})}{\mathbb{E}} \left[ \sum_{I \subseteq [n]: f_I \neq 0, i \in I} \underset{\pi \sim U(\Omega)}{\mathbb{E}} \left[ f_I(r_{\pi_{:i} \cup \{i\}}(z, x)_I) \right] - \underset{\pi \sim U(\Omega)}{\mathbb{E}} \left[ f_I(r_{\pi_{:i}}(z, x)_I) \right] \right]. \tag{31}$$

Lemma A.1 gives the value of $\mathbb{E}_{\pi \sim U(\Omega)} \left[ f_I(r_{\pi_{:i} \cup \{i\}}(z, x)_I) \right]$, and Lemma A.2 gives the value of $\mathbb{E}_{\pi \sim U(\Omega)} \left[ f_I(r_{\pi_{:i}}(z, x)_I) \right]$. □

**Lemma A.1.** *Consider the definitions of Section 2, assuming that $i \in I$, and the following:*

$$\mathcal{A}^+(i, I) := \{A \subset I | i \in A, 1 \le |A| < |I|\}. \tag{32}$$

*In other words, $\mathcal{A}^+(i, I)$ is the family of strict subsets of $I$ which contain $i$. Then we have:*

$$\mathbb{E}_{\pi \sim U(\Omega)} \left[ f_I(r_{\pi_{:i} \cup \{i\}}(z, x)_I) \right] = \frac{f_I(x_I)}{|I|} + \sum_{A \in \mathcal{A}^+(i, I)} \frac{f_I(r_A(z, x)_I)}{|A| \binom{|I|}{|A|}}. \tag{33}$$

**Lemma A.2.** *Consider the definitions of Section 2, assuming that $i \in I$, as well as:*

$$\mathcal{A}^-(i, I) := \{A \subseteq I | i \in A, 1 < |A| \le |I|\}. \tag{34}$$

*In other words, $\mathcal{A}^-(i, I)$ is the family of subsets of $I$ which contain $i$ and at least one other element. Then we have:*

$$\mathbb{E}_{\pi \sim U(\Omega)} \left[ f_I(r_{\pi_{:i}}(z, x)_I) \right] = \frac{f_I(z_I)}{|I|} + \sum_{A \in \mathcal{A}^-(i, I)} \frac{f_I(r_{A \setminus \{i\}}(z, x)_I)}{|A| \binom{|I|}{|A|}}. \tag{35}$$

*Proof of Lemma A.1.* We can separate the expectation into two parts:

$$\mathbb{E}_{\pi \sim U(\Omega)} \left[ f_I(r_{\pi_{:i} \cup \{i\}}(z, x)_I) \right] = \underbrace{\frac{1}{n!} \sum_{\substack{\pi \in \Omega \\ I \subseteq \pi_{:i} \cup \{i\}}} f_I(r_{\pi_{:i} \cup \{i\}}(z, x)_I)}_{\text{Term 1}} + \underbrace{\frac{1}{n!} \sum_{\substack{\pi \in \Omega \\ I \setminus (\pi_{:i} \cup \{i\}) \ne \emptyset}} f_I(r_{\pi_{:i} \cup \{i\}}(z, x)_I)}_{\text{Term 2}}. \tag{36}$$

**Calculating Term 1.** The key to calculating the first half of the right-hand side of the previous equation, which we refer to as Term 1, is to notice that $r_{\pi_{:i} \cup \{i\}}(z, x)_I = x_I$ for all the permutations $\pi$ that satisfy $I \subseteq \pi_{:i} \cup \{i\}$, and so the value $f_I(r_{\pi_{:i} \cup \{i\}}(z, x)_I)$ is constant over all those permutations. It is simply $f_I(r_{\pi_{:i} \cup \{i\}}(z, x)_I) = f_I(x_I)$. All that we need to do is calculate the number of such permutations. There is an elegant combinatorics argument that gives us the solution immediately.

Consider any permutation $\pi$ of $[n]$. This permutation satisfies the condition $I \subseteq \pi_{:i} \cup \{i\}$ if and only if $i$ is to the right of every other variable of $I$. This rightmost position is one possibility of out of $|I|$. This means that exactly 1 out of every $|I|$ permutations satisfies the condition. The number of permutations we seek is $\frac{n!}{|I|}$. We therefore have:

$$\text{Term 1} = \frac{1}{n!} \frac{n!}{|I|} f_I(x_I) = \frac{f_I(x_I)}{|I|}. \tag{37}$$

**Calculating Term 2.** Here, we consider all the permutations $\pi$ such that $I \setminus (\pi_{:i} \cup \{i\}) \ne \emptyset$, i.e. the feature subset $I$ contains variables not in $\pi_{:i} \cup \{i\}$. These variables will be replaced when calculating the function. Defining $A := I \cap (\pi_{:i} \cup \{i\})$, the set of shared variables (note that we always have $i \in A$), we have:

$$r_{\pi_{:i} \cup \{i\}}(z, x)_I = r_A(z, x)_I.$$

The function $f_I(r_{\pi_{:i} \cup \{i\}}(z, x)_I)$ therefore takes a different value for each possible $A$, of which there are:

$$\begin{aligned}
|\mathcal{A}^+(i, I)| &= \sum_{k=0}^{|I|-2} \binom{|I|-1}{k} \\
&= \sum_{k=0}^{|I|-1} \binom{|I|-1}{k} - \binom{|I|-1}{|I|-1} \\
&= 2^{|I|-1} - 1.
\end{aligned} \tag{38}$$

(We can choose 0 up to $|I| - 2$ variables to accompany $i$ in making up $A \in \mathcal{A}^+(i, I)$.) For a given set $A \in \mathcal{A}^+(i, I)$ of shared variables, we must calculate the number of permutations $\pi$ that are such that $A = I \cap (\pi_{:i} \cup \{i\})$. Call that number $N(n, |I|, |A|)$. We have:

$$\text{Term 2} = \sum_{A \in \mathcal{A}^+(i, I)} \frac{N(n, |I|, |A|)}{n!} f_I(r_A(z, x)_I). \tag{39}$$

Let us now calculate $N(n, |I|, |A|)$.

1. First, choose the $|I|$ positions for the variables in $I$. There are $\binom{n}{|I|}$ possibilities.

2. Of these $|I|$ positions, there is only one choice for $i$ itself, since the variables in $A \setminus \{i\}$ must be to the left of it, and the variables in $I \setminus A$ to the right.

3. We can permute the $|A| - 1$ variables of $A \setminus \{i\}$, the $|I| - |A|$ variables of $I \setminus A$, and the $n - |I|$ variables in $[n] \setminus I$, for a total of $(|A| - 1)!(|I| - |A|)!(n - |I|)!$ permutations.

Assembling these facts gives us the formula:

$$N(n, |I|, |A|) := (|A| - 1)!(|I| - |A|)!(n - |I|)! \binom{n}{|I|} \tag{40}$$

We can simplify this further by expanding the binomial coefficient:

$$\begin{aligned}
N(n, |I|, |A|) &= (|A| - 1)!(|I| - |A|)!(n - |I|)! \binom{n}{|I|} \\
&= \frac{n!(|A| - 1)!(|I| - |A|)!(n - |I|)!}{|I|!(n - |I|)!} \\
&= \frac{n!}{|A| \binom{|I|}{|A|}}.
\end{aligned} \tag{41}$$

This gives us the result. □

*Proof of Lemma A.2.* The proof is quite similar to that of Lemma A.1. We can separate the expectation into two parts:

$$\mathop{\mathbb{E}}_{\pi \sim U(\Omega)} [f_I(r_{\pi_{:i}}(z, x)_I)] = \underbrace{\frac{1}{n!} \sum_{\substack{\pi \in \Omega \\ I \cap \pi_{:i} = \emptyset}} f_I(r_{\pi_{:i}}(z, x)_I)}_{\text{Term 1}} + \underbrace{\frac{1}{n!} \sum_{\substack{\pi \in \Omega \\ I \cap \pi_{:i} \neq \emptyset}} f_I(r_{\pi_{:i}}(z, x)_I)}_{\text{Term 2}}. \tag{42}$$

**Calculating Term 1.** Notice that $r_{\pi_{:i}}(z, x)_I = z_I$ for all the permutations $\pi$ that satisfy $I \cap \pi_{:i} = \emptyset$, and so the value $f_I(r_{\pi_{:i}}(z, x)_I)$ is constant over all those permutations. It is simply $f_I(r_{\pi_{:i}}(z, x)_I) = f_I(z_I)$. All that we need to do to calculate the number of such permutations. However, a simple symmetry argument gives us the answer. In the proof of Lemma A.1, we showed that the number of permutations $\pi$ such that $I \subseteq \pi_{:i} \cup \{i\}$ is $\frac{n!}{|I|}$. In fact, we can notice that this number should be the same as the number of permutations that satisfy $I \cap \pi_{:i} = \emptyset$. Indeed, in the first case, all variables of $I \setminus \{i\}$ must be to the left of $i$ in $\pi_{:i}$. In the second, they must be to the right. There are exactly as many permutations that satisfy each condition.

**Calculating Term 2.** Here, we consider all the permutations $\pi$ such that $I \cap \pi_{:i} \neq \emptyset$, i.e. the feature subset $I$ shares at least one variable with $\pi_{:i}$. In this situation, some or all variables are replaced when calculating the function. Using the same definition $A := I \cap (\pi_{:i} \cup \{i\})$ as in the proof of Lemma A.1, we have:

$$r_{\pi_{:i}}(z, x)_I = r_{A \setminus \{i\}}(z, x)_I.$$

The function $f_I(r_{\pi_{:i}}(z,x)_I)$ therefore takes a different value for each possible $A$. Since each $A$ must contain at least two elements ($i$, and one more variable so that $I \cap \pi_{:i} \neq \emptyset$), and also $A = I$ is now acceptable, the intersection $A$ is taken from the set $\mathcal{A}^-(i,I)$, which has the same cardinality as the set $\mathcal{A}^+(i,I)$, namely $2^{|I|-1} - 1$. We also know from the previous proof that there are $N(n,|I|,|A|) = \frac{n!}{|A|\binom{|I|}{|A|}}$ permutations $\pi$ such that $A = I \cap (\pi_{:i} \cup \{i\})$. We therefore have:

$$\text{Term 2} = \sum_{A \in \mathcal{A}^-(i,I)} \frac{f_I(r_{A\setminus\{i\}}(z,x)_I)}{|A|\binom{|I|}{|A|}}, \tag{43}$$

which concludes the proof. $\qquad\square$

## B Calculus for RWRelu expectation

Here we calculate the expectation $\mathbb{E}_{w \sim p}[\mathcal{K}(u,w)\phi(w,x)]$ for RWRelu (required to calculate the output of the model) through a series of lemmas. First, we recall a pair of lemmas from Dubé & Marchand (2024).

**Lemma B.1** (Lemma 15 of Dubé & Marchand (2024)). *Consider a Hilbert space $\mathcal{W}$. Let $u, w \in \mathcal{W}$ and $a, b > 0$. Then:*

$$\frac{\|w - u\|^2}{a} + \frac{\|w\|^2}{b} = \left(\frac{1}{a} + \frac{1}{b}\right)\left\|w - \frac{1}{1 + \frac{a}{b}}u\right\|^2 + \frac{1}{a+b}\|u\|^2.$$

**Lemma B.2** (Lemma 16 of Dubé & Marchand (2024)). *We have:*

$$\int_{\mathbb{R}^n} e^{-a\|w-u\|^2} \mathrm{d}w = \left(\frac{\pi}{a}\right)^{\frac{n}{2}}.$$

**Lemma B.3.** *We have:*

$$\int_{-\infty}^{\infty} e^{-(w-u)^2/2\gamma^2} \max(0, wx)\mathrm{d}w = \frac{\gamma|x|}{\sqrt{2}}\left(\sqrt{2}\gamma e^{-u^2/2\gamma^2} + \sqrt{\pi}u\left(\text{sign}(x) + \text{erf}\left(\frac{u}{\sqrt{2}\gamma}\right)\right)\right).$$

*Proof.* We have:

$$\int_{-\infty}^{\infty} e^{-(w-u)^2/2\gamma^2} \max(0, wx)\mathrm{d}w = \int_{-\infty}^{\infty} e^{-t^2} \max\left(0, \left(\sqrt{2}\gamma t + u\right)x\right)\sqrt{2}\gamma\mathrm{d}w \qquad (t := \tfrac{w-u}{\sqrt{2}\gamma}, \ \mathrm{d}t = \tfrac{\mathrm{d}w}{\sqrt{2}\gamma})$$

$$= \sqrt{2}\gamma \int_{-\infty}^{\infty} e^{-t^2} \max\left(0, \left(\sqrt{2}\gamma t + u\right)x\right)\mathrm{d}t.$$

**Case 1.** If $x \geq 0$, we have:

$$\sqrt{2}\gamma \int_{-\infty}^{\infty} e^{-t^2} \max\left(0, \left(\sqrt{2}\gamma t + u\right)x\right)\mathrm{d}t$$

$$= \sqrt{2}\gamma \int_{-u/\sqrt{2}\gamma}^{\infty} e^{-t^2} \left(\sqrt{2}\gamma t + u\right)x\,\mathrm{d}t$$

$$= \sqrt{2}\gamma x\left(\sqrt{2}\gamma \int_{-u/\sqrt{2}\gamma}^{\infty} te^{-t^2}\mathrm{d}t + u \int_{-u/\sqrt{2}\gamma}^{\infty} e^{-t^2}\mathrm{d}t\right)$$

$$= \sqrt{2}\gamma x\left(\sqrt{2}\gamma \left(\frac{-e^{-t^2}}{2}\right)\Big|_{t=-u/\sqrt{2}\gamma}^{t=\infty} + u\left[\int_{-u/\sqrt{2}\gamma}^{0} e^{-t^2}\mathrm{d}t + \int_0^{\infty} e^{-t^2}\mathrm{d}t\right]\right)$$

$$= \sqrt{2}\gamma x\left(\frac{\gamma}{\sqrt{2}}e^{-u^2/2\gamma^2} + u\left[\frac{\sqrt{\pi}}{2}\text{erf}\left(\frac{u}{\sqrt{2}\gamma}\right) + \frac{\sqrt{\pi}}{2}\text{erf}(\infty)\right]\right)$$

$$= \frac{\gamma x}{\sqrt{2}}\left(\sqrt{2}\gamma e^{-u^2/2\gamma^2} + \sqrt{\pi}u\left[1 + \text{erf}\left(\frac{u}{\sqrt{2}\gamma}\right)\right]\right).$$

**Case 2.** If $x < 0$, we have instead:

$$\sqrt{2}\gamma \int_{-\infty}^{\infty} e^{-t^2} \max\left(0, \left(\sqrt{2}\gamma t + u\right)x\right) \mathrm{d}t$$

$$= \sqrt{2}\gamma \int_{-\infty}^{-u/\sqrt{2}\gamma} e^{-t^2} \left(\sqrt{2}\gamma t + u\right)x \mathrm{d}t$$

$$= \sqrt{2}\gamma x \left(\sqrt{2}\gamma \int_{-\infty}^{-u/\sqrt{2}\gamma} te^{-t^2} \mathrm{d}t + u \int_{-\infty}^{-u/\sqrt{2}\gamma} e^{-t^2} \mathrm{d}t\right)$$

$$= \sqrt{2}\gamma x \left(\sqrt{2}\gamma \left(\frac{-e^{-t^2}}{2}\Bigg|_{t=-\infty}^{t=-u/\sqrt{2}\gamma}\right) + u\left[\int_{-\infty}^{0} e^{-t^2}\mathrm{d}t - \int_{-u/\sqrt{2}\gamma}^{0} e^{-t^2}\mathrm{d}t\right]\right)$$

$$= \sqrt{2}\gamma x \left(\frac{-\gamma}{\sqrt{2}}e^{-u^2/2\gamma^2} + u\left[\frac{\sqrt{\pi}}{2}\operatorname{erf}(\infty) - \frac{\sqrt{\pi}}{2}\operatorname{erf}\left(\frac{u}{\sqrt{2}\gamma}\right)\right]\right)$$

$$= \frac{\gamma x}{\sqrt{2}}\left(-\sqrt{2}\gamma e^{-u^2/2\gamma^2} + \sqrt{\pi}u\left[1 - \operatorname{erf}\left(\frac{u}{\sqrt{2}\gamma}\right)\right]\right).$$

Both cases can be combined using the sign of $x$, leading to the desired result. $\qquad\square$

**Lemma B.4.** *We have*

$$\int_{\mathbb{R}^n} e^{-\|w-u\|^2/2\gamma^2} \max(0, \langle w, x \rangle) \mathrm{d}w$$

$$= \left(\sqrt{2\pi\gamma^2}\right)^{n-1}\left[\frac{\gamma\|x\|}{\sqrt{2}}\left(\sqrt{2}\gamma e^{-\frac{\langle u,x\rangle^2}{2\gamma^2\|x\|^2}} + \sqrt{\pi}\frac{\langle u,x\rangle}{\|x\|}\left[1 + \operatorname{erf}\left(\frac{\langle u,x\rangle}{\sqrt{2}\gamma\|x\|}\right)\right]\right)\right].$$

*Proof.* Calculate the integral using an orthonormal basis $\{v_1, \ldots, v_n\}$ of $\mathbb{R}^n$ such that $v_n := \frac{x}{\|x\|}$. Write $w = (w_1, \ldots, w_n)$ in this new basis (i.e. $w_i := \langle w, v_i \rangle$ for all $i$), and similarly $(u_1, \ldots, u_n)$ for $u$. Under this change of coordinates, the integral becomes:

$$\int_{\mathbb{R}^n} e^{-\|w-u\|^2/2\gamma^2} \max(0, \langle w, x \rangle) \mathrm{d}w$$

$$= \int_{\mathbb{R}} \int_{\mathbb{R}^{n-1}} e^{-\left[\sum_{i=1}^{n-1}(w_i-u_i)^2 + (w_n-u_n)^2\right]/2\gamma^2} \max(0, u_n\|x\|) \mathrm{d}w_1 \ldots \mathrm{d}w_n.$$

We are left with a product of $n$ independent integrals:

$$\int_{\mathbb{R}^n} e^{-\|w-u\|^2/2\gamma^2} \max(0, \langle w, x \rangle) \mathrm{d}w$$

$$= \int_{\mathbb{R}^{n-1}} e^{-\sum_{i=1}^{n-1}(w_i-u_i)^2/2\gamma^2} \mathrm{d}w_1 \ldots \mathrm{d}w_{n-1} \int_{\mathbb{R}} e^{-(w_n-u_n)^2/2\gamma^2} \max(0, w_n\|x\|) \mathrm{d}w_n$$

$$= \int_{\mathbb{R}^{n-1}} \prod_{i=1}^{n-1} e^{-(w_i-u_i)^2/2\gamma^2} \mathrm{d}w_1 \ldots \mathrm{d}w_{n-1} \int_{\mathbb{R}} e^{-(w_n-u_n)^2/2\gamma^2} \max(0, w_n\|x\|) \mathrm{d}w_n$$

$$= \prod_{i=1}^{n-1} \int_{-\infty}^{\infty} e^{-(w_i-u_i)^2/2\gamma^2} \mathrm{d}w_i \int_{\mathbb{R}} e^{-(w_n-u_n)^2/2\gamma^2} \max(0, w_n\|x\|) \mathrm{d}w_n.$$

For each $i$, Lemma B.2 gives us:

$$\int_{\mathbb{R}} e^{-(w_i-u_i)^2/2\gamma^2} \mathrm{d}w_i = \sqrt{2\pi\gamma^2}.$$

Also, Lemma B.3 gives us:

$$\int_{\mathbb{R}} e^{-(w_n - u_n)^2/2\gamma^2} \max(0, w_n \|x\|) \mathrm{d}w_n$$
$$= \frac{\gamma \|x\|}{\sqrt{2}} \left( \sqrt{2}\gamma e^{-u_n^2/2\gamma^2} + \sqrt{\pi} u_n \left( \operatorname{sign}(\|x\|) + \operatorname{erf}\left( \frac{u_n}{\sqrt{2}\gamma} \right) \right) \right).$$

Finally, since $u_n = \frac{\langle u, x \rangle}{\|x\|}$ and $\operatorname{sign}(\|x\|)$, we have the result. $\qquad\square$

*Proof of Theorem 4.1.* We have:

$$\mathbb{E}_{w \sim p} [\mathcal{K}(u, w)\phi(w, x)] = \left( \frac{1}{\sqrt{2\pi\sigma^2}} \right)^n \int_{\mathbb{R}^n} e^{-\|w - u\|^2/2\gamma^2} e^{-\|w\|^2/2\sigma^2} \max(0, \langle w, x \rangle) \mathrm{d}w$$

$$= \left( \frac{1}{\sqrt{2\pi\sigma^2}} \right)^n e^{\frac{-\|u\|_2^2}{2\sigma^2 + 2\gamma^2}} \int_{\mathbb{R}^n} e^{-\left( \frac{1}{2\gamma^2} + \frac{1}{2\sigma^2} \right) \left\| w - \frac{u}{1 + \frac{\gamma^2}{\sigma^2}} \right\|^2} \max(0, \langle w, x \rangle) \mathrm{d}w \quad \text{(Lemma B.1)}$$

$$= \left( \frac{1}{\sqrt{2\pi\sigma^2}} \right)^n e^{\frac{-\|u\|_2^2}{2\sigma^2 + 2\gamma^2}} \int_{\mathbb{R}^n} e^{-\|w - u'\|^2/2\zeta^2} \max(0, \langle w, x \rangle) \mathrm{d}w.$$

Applying Lemma B.4, we obtain:

$$\mathbb{E}_{w \sim p} [\mathcal{K}(u, w)\phi(w, x)]$$

$$= \left( \frac{1}{\sqrt{2\pi\sigma^2}} \right)^n e^{\frac{-\|u\|_2^2}{2\sigma^2 + 2\gamma^2}} \left( \sqrt{2\pi\zeta^2} \right)^{n-1} \left[ \frac{\zeta \|x\|}{\sqrt{2}} \left( \sqrt{2}\zeta e^{-\frac{\langle u', x \rangle^2}{2\zeta^2 \|x\|^2}} + \sqrt{\pi} \frac{\langle u', x \rangle}{\|x\|} \left[ 1 + \operatorname{erf}\left( \frac{\langle u', x \rangle}{\sqrt{2}\zeta \|x\|} \right) \right] \right) \right]$$

$$= \left( 1 + \frac{\sigma^2}{\gamma^2} \right)^{-n/2} e^{\frac{-\|u\|_2^2}{2\sigma^2 + 2\gamma^2}} \frac{\|x\|}{2\sqrt{\pi}} \left( \sqrt{2}\zeta e^{-\frac{\langle u', x \rangle^2}{2\zeta^2 \|x\|^2}} + \sqrt{\pi} \frac{\langle u', x \rangle}{\|x\|} \left[ 1 + \operatorname{erf}\left( \frac{\langle u', x \rangle}{\sqrt{2}\zeta \|x\|} \right) \right] \right).$$

$$\square$$

## C   Theoretical constants of RWRelu

For each of the two instantiations Dubé & Marchand (2024) introduce, they provide the value of two theoretical constants, $\kappa$ and $\theta$, relevant for their theoretical guarantees. For a given instantiation $(\mathcal{W}, \phi, \mathcal{K}, p)$, we have:

$$\kappa := \sup_{x \in \mathcal{X}} \sqrt{\mathbb{E}_{w \sim p} \left[ \|\phi(w, x)\mathcal{K}(w, \cdot)\|_{\mathcal{H}}^2 \right]} = \sup_{x \in \mathcal{X}} \sqrt{\mathbb{E}_{w \sim p} [\mathcal{K}(w, w)\phi(w, x)^2]} \tag{44}$$

$$\theta := \sup_{x \in \mathcal{X}} \left\| \mathbb{E}_{w \sim p} [\phi(w, x)\mathcal{K}(w, \cdot)] \right\|_{\mathcal{H}} = \sup_{x \in \mathcal{X}} \sqrt{\mathbb{E}_{w \sim p} \mathbb{E}_{u \sim p} [\mathcal{K}(u, w)\phi(u, x)\phi(w, x)]}. \tag{45}$$

For completeness, we give here the exact value of $\kappa$ and an upper bound for $\theta$, and the relevant calculus, in the case of instantiation RWRelu. We begin by a couple of lemmas to help with the calculus.

**Lemma C.1.** *Consider $\sigma > 0$ and $\gamma > 0$. Then:*

$$\int_{\mathbb{R}^n} \int_{\mathbb{R}^n} e^{-\|u - w\|^2/2\gamma^2} e^{-\langle u, w \rangle/\sigma^2} \mathrm{d}u \mathrm{d}w = (2\pi)^n \left( \frac{\gamma^2 \sigma^4}{2\sigma^2 - \gamma^2} \right)^{n/2}. \tag{46}$$

*Proof.*

$$\int_{\mathbb{R}^n}\int_{\mathbb{R}^n} e^{-\frac{\|u-w\|^2}{2\gamma^2}} e^{-\frac{\langle u,w\rangle}{\sigma^2}}\,\mathrm{d}u\mathrm{d}w = \int_{\mathbb{R}^n}\int_{\mathbb{R}^n} e^{-\frac{\|t\|^2}{2\gamma^2}} e^{-\frac{\langle t+w,w\rangle}{\sigma^2}}\,\mathrm{d}t\mathrm{d}w \qquad (t := u - w,\ \mathrm{d}t = \mathrm{d}u)$$

$$= \int_{\mathbb{R}^n}\int_{\mathbb{R}^n} e^{-\frac{\|t\|^2}{2\gamma^2}} e^{-\frac{\|w\|^2}{\sigma^2}} e^{-\frac{\langle t,w\rangle}{\sigma^2}}\,\mathrm{d}t\mathrm{d}w$$

$$= \int_{\mathbb{R}^n} e^{-\frac{\|w\|^2}{\sigma^2}} \left[\int_{\mathbb{R}^n} e^{-\frac{\|t\|^2}{2\gamma^2}} e^{-\frac{\langle t,w\rangle}{\sigma^2}}\,\mathrm{d}t\right]\mathrm{d}w$$

$$= \int_{\mathbb{R}^n} e^{-\frac{\|w\|^2}{\sigma^2}} \left[\int_{\mathbb{R}^n} e^{-\frac{\left\|t+\frac{\gamma^2 w}{\sigma^2}\right\|^2}{2\gamma^2}} e^{\frac{\left\|\frac{\gamma^2 w}{\sigma^2}\right\|^2}{2\gamma^2}}\,\mathrm{d}t\right]\mathrm{d}w$$

$$= \int_{\mathbb{R}^n} e^{-\frac{\|w\|^2}{\sigma^2}} e^{\frac{\left\|\frac{\gamma^2 w}{\sigma^2}\right\|^2}{2\gamma^2}} \left[\int_{\mathbb{R}^n} e^{-\frac{\left\|t+\frac{\gamma^2 w}{\sigma^2}\right\|^2}{2\gamma^2}}\,\mathrm{d}t\right]\mathrm{d}w$$

$$= \left(\sqrt{2\pi\gamma^2}\right)^n \int_{\mathbb{R}^n} e^{-\frac{\|w\|^2}{\sigma^2}} e^{\frac{\left\|\frac{\gamma^2 w}{\sigma^2}\right\|^2}{2\gamma^2}}\,\mathrm{d}w.$$

Then, simplifying the exponent:

$$-\frac{\|w\|^2}{\sigma^2} + \frac{\left\|\frac{\gamma^2 w}{\sigma^2}\right\|^2}{2\gamma^2} = -\frac{\|w\|^2}{2\sigma^2}\left(2 - \frac{\gamma^2}{\sigma^2}\right)$$

$$= -\frac{\|w\|^2}{2\sigma^2}\left(\frac{2\sigma^2 - \gamma^2}{\sigma^2}\right)$$

$$= -\frac{\|w\|^2}{2\sigma^4}\left(2\sigma^2 - \gamma^2\right),$$

we get:

$$\int_{\mathbb{R}^n}\int_{\mathbb{R}^n} e^{-\frac{\|u-w\|^2}{2\gamma^2}} e^{-\frac{\langle u,w\rangle}{\sigma^2}}\,\mathrm{d}u\mathrm{d}w = \left(\sqrt{2\pi\gamma^2}\right)^n \int_{\mathbb{R}^n} e^{-\frac{\|w\|^2}{2\sigma^4}\left(2\sigma^2-\gamma^2\right)}\,\mathrm{d}w$$

$$= \left(\sqrt{2\pi\gamma^2}\right)^n \left(\sqrt{2\pi\frac{\sigma^4}{2\sigma^2-\gamma^2}}\right)^n$$

$$= (2\pi)^n \left(\frac{\gamma^2\sigma^4}{2\sigma^2-\gamma^2}\right)^{n/2}.$$

$\square$

**Lemma C.2.** *Consider $\sigma > 0$ and $\gamma > 0$. Denote $I_n$ the identity matrix in $\mathbb{R}^n$. Then:*

$$\mathbb{E}_{w\sim\mathcal{N}(0,\sigma^2 I_n)}\ \mathbb{E}_{u\sim\mathcal{N}(0,\sigma^2 I_n)}\left[e^{-\|u-w\|^2/2\gamma^2}\right] = \left(1 + \frac{2\sigma^2}{\gamma^2}\right)^{-n/2}. \tag{47}$$

*Proof.* The expectation is a straightforward integral:

$$
\begin{aligned}
\mathbb{E}_{w\sim p}\mathbb{E}_{u\sim p}[\mathcal{K}(u,w)] &= \left(\frac{1}{\sqrt{2\pi\sigma^2}}\right)^{2n}\int_{\mathbb{R}^n}\int_{\mathbb{R}^n} e^{-\frac{\|u-w\|^2}{2\gamma^2}} e^{-\frac{\|u\|^2}{2\sigma^2}} e^{-\frac{\|w\|^2}{2\sigma^2}}\,\mathrm{d}u\mathrm{d}w \\
&= \left(\frac{1}{\sqrt{2\pi\sigma^2}}\right)^{2n}\int_{\mathbb{R}^n}\int_{\mathbb{R}^n} e^{-\frac{\|u-w\|^2}{2\gamma^2}} e^{-\frac{\|u\|^2}{2\sigma^2}} e^{\frac{\langle u,w\rangle}{\sigma^2}} e^{-\frac{\|w\|^2}{2\sigma^2}} e^{-\frac{\langle u,w\rangle}{\sigma^2}}\,\mathrm{d}u\mathrm{d}w \\
&= \left(\frac{1}{\sqrt{2\pi\sigma^2}}\right)^{2n}\int_{\mathbb{R}^n}\int_{\mathbb{R}^n} e^{-\frac{\|u-w\|^2}{2\gamma^2}} e^{-\frac{\|u-w\|^2}{2\sigma^2}} e^{-\frac{\langle u,w\rangle}{\sigma^2}}\,\mathrm{d}u\mathrm{d}w \\
&= \left(\frac{1}{\sqrt{2\pi\sigma^2}}\right)^{2n}\int_{\mathbb{R}^n}\int_{\mathbb{R}^n} e^{-\frac{\|u-w\|^2}{2\zeta^2}} e^{-\frac{\langle u,w\rangle}{\sigma^2}}\,\mathrm{d}u\mathrm{d}w \qquad \left(\tfrac{1}{2\zeta^2}=\tfrac{1}{2\gamma^2}+\tfrac{1}{2\sigma^2}=\tfrac{\sigma^2+\gamma^2}{2\sigma^2\gamma^2}\right) \\
&= \left(\frac{1}{\sqrt{2\pi\sigma^2}}\right)^{2n}(2\pi)^n\left(\frac{\zeta^2\sigma^4}{2\sigma^2-\zeta^2}\right)^{n/2} \qquad\qquad\qquad\text{(Lemma C.1)} \\
&= \left(\frac{\zeta^2}{2\sigma^2-\zeta^2}\right)^{n/2} \\
&= \left(\frac{2\sigma^2}{\zeta^2}-1\right)^{-n/2} \\
&= \left(\frac{2\sigma^2}{\sigma^2}+\frac{2\sigma^2}{\gamma^2}-1\right)^{-n/2} \\
&= \left(1+\frac{2\sigma^2}{\gamma^2}\right)^{-n/2}.
\end{aligned}
$$

$\square$

**Lemma C.3.** *Considering RWRelu, we have:*

$$
\kappa = \frac{\sigma}{\sqrt{2}}\sup_{x\in\mathcal{X}}\|x\|.
$$

*Proof.* We have:

$$
\begin{aligned}
\kappa^2 &:= \sup_{x\in\mathcal{X}}\mathbb{E}_{w\sim p}\left[\mathcal{K}(w,w)\phi(w,x)^2\right] \\
&= \sup_{x\in\mathcal{X}}\mathbb{E}_{w\sim p}\left[\max\left(0,\langle w,x\rangle\right)^2\right] \\
&= \sup_{x\in\mathcal{X}}\left(\frac{1}{\sqrt{2\pi\sigma^2}}\right)^n\int_{\mathbb{R}^n}e^{-\|w\|^2/2\sigma^2}\max\left(0,\langle w,x\rangle\right)^2\mathrm{d}w \\
&= \sup_{x\in\mathcal{X}}\frac{1}{\sqrt{2\pi\sigma^2}}\int_{\mathbb{R}}e^{-w_n^2/2\sigma^2}\max(0,w_n\|x\|)^2\mathrm{d}w_n \qquad\left(w_n:=\tfrac{\langle w,x\rangle}{\|x\|},\text{ see proof of Lemma B.4}\right) \\
&= \sup_{x\in\mathcal{X}}\frac{1}{\sqrt{2\pi\sigma^2}}\int_0^\infty e^{-w_n^2/2\sigma^2}w_n^2\|x\|^2\mathrm{d}w_n \\
&= \sup_{x\in\mathcal{X}}\|x\|^2\frac{1}{\sqrt{2\pi\sigma^2}}\int_0^\infty e^{-w_n^2/2\sigma^2}w_n^2\mathrm{d}w_n \\
&= \frac{1}{2}\sup_{x\in\mathcal{X}}\|x\|^2\mathbb{E}_{w_n\sim\mathcal{N}(0,\sigma^2)}\left[w_n^2\right] \\
&= \frac{\sigma^2}{2}\sup_{x\in\mathcal{X}}\|x\|^2.
\end{aligned}
$$

We have the result by taking the square root. $\square$

**Lemma C.4.** *Considering RWRelu, we have:*

$$\theta^2 \leq \sup_{x \in \mathcal{X}} \|x\|^2 \left(1 + \frac{2\sigma^2}{\gamma^2}\right)^{-(n-1)/2} \left(\frac{\sigma^2}{2\pi}\right). \tag{48}$$

*Proof.* We have:

$$\theta^2 := \sup_{x \in \mathcal{X}} \underset{u \sim p}{\mathbb{E}} \underset{w \sim p}{\mathbb{E}} \left[\mathcal{K}(u, w)\phi(u, x)\phi(w, x)\right]$$

$$= \sup_{x \in \mathcal{X}} \left(\frac{1}{2\pi\sigma^2}\right)^n \int_{\mathbb{R}^n} \int_{\mathbb{R}^n} e^{-\|u-w\|^2/2\gamma^2} e^{-\|u\|^2/2\sigma^2} e^{-\|w\|^2/2\sigma^2} \max(0, \langle u, x\rangle) \max(0, \langle w, x\rangle) \mathrm{d}u \mathrm{d}w.$$

We are free to solve these integrals using any orthonormal basis of $\mathbb{R}^n$. We choose a basis $\{v_1, \ldots, v_n\}$ such that:

$$v_n := \frac{x}{\|x\|}. \tag{49}$$

In particular, we have:

$$u_n := \left\langle u, \frac{x}{\|x\|}\right\rangle,$$

$$w_n := \left\langle w, \frac{x}{\|x\|}\right\rangle,$$

$$\langle u, x\rangle = u_n\|x\|,$$

$$\langle w, x\rangle = w_n\|x\|.$$

Denoting $u_{1:n-1} := (u_1, \ldots, u_{n-1})$ and $w_{1:n-1} := (w_1, \ldots, w_{n-1})$, we can rewrite the expression:

$$e^{-\|u-w\|^2/2\gamma^2} e^{-\|u\|^2/2\sigma^2} e^{-\|w\|^2/2\sigma^2} \tag{50}$$

as:

$$e^{-\|u_{1:n-1}-w_{1:n-1}\|^2/2\gamma^2} e^{-\|u_{1:n-1}\|^2/2\sigma^2} e^{-\|w_{1:n-1}\|^2/2\sigma^2} e^{-(u_n-w_n)^2/2\gamma^2} e^{-u_n^2/2\sigma^2} e^{-w_n^2/2\sigma^2}. \tag{51}$$

This allows us to separate the integral into two parts. The first one is the integral over $(u_1, \ldots, u_{n-1})$ and $(w_1, \ldots, w_{n-1})$:

$$\left(\frac{1}{2\pi\sigma^2}\right)^{n-1} \int_{\mathbb{R}^{n-1}} \int_{\mathbb{R}^{n-1}} e^{-\|u_{1:n-1}-w_{1:n-1}\|^2/2\gamma^2} e^{-\|u_{1:n-1}\|^2/2\sigma^2} e^{-\|w_{1:n-1}\|^2/2\sigma^2} \mathrm{d}u_{1:n-1} \mathrm{d}w_{1:n-1}. \tag{52}$$

Lemma C.2 tells us that the previous expression is equal to:

$$\left(1 + \frac{2\sigma^2}{\gamma^2}\right)^{-(n-1)/2}. \tag{53}$$

The second integral is the one over $u_n$ and $w_n$:

$$\left(\frac{1}{2\pi\sigma^2}\right) \int_{\mathbb{R}} \int_{\mathbb{R}} e^{-(u_n-w_n)^2/2\gamma^2} e^{-u_n^2/2\sigma^2} e^{-w_n^2/2\sigma^2} \max(0, u_n\|x\|) \max(0, w_n\|x\|) \mathrm{d}u_n \mathrm{d}w_n. \tag{54}$$

We can simplify the integral by noticing that $\max(0, u_n) = u_n$ if $u_n \geq 0$, and 0 otherwise:

$$\left(\frac{1}{2\pi\sigma^2}\right) \|x\|^2 \int_0^\infty \int_0^\infty e^{-(u_n-w_n)^2/2\gamma^2} e^{-u_n^2/2\sigma^2} e^{-w_n^2/2\sigma^2} u_n w_n \mathrm{d}u_n \mathrm{d}w_n. \tag{55}$$

The expression for $\theta^2$ has now become:

$$\theta^2 = \sup_{x \in \mathcal{X}} \left(\frac{1}{2\pi\sigma^2}\right)^n \int_{\mathbb{R}^n} \int_{\mathbb{R}^n} e^{-\|u-w\|^2/2\gamma^2} e^{-\|u\|^2/2\sigma^2} e^{-\|w\|^2/2\sigma^2} \max(0, \langle u, x\rangle) \max(0, \langle w, x\rangle) \mathrm{d}u \mathrm{d}w$$

$$= \sup_{x \in \mathcal{X}} \|x\|^2 \left(1 + \frac{2\sigma^2}{\gamma^2}\right)^{-(n-1)/2} \left(\frac{1}{2\pi\sigma^2}\right) \int_0^\infty \int_0^\infty e^{-(u_n-w_n)^2/2\gamma^2} e^{-u_n^2/2\sigma^2} e^{-w_n^2/2\sigma^2} u_n w_n \mathrm{d}u_n \mathrm{d}w_n. \tag{56}$$

The remaining integral is difficult. We can simplify by using the fact that $e^{-(u_n - w_n)^2/2\gamma^2} \leq 1$, and obtain an upper bound on $\theta^2$:

$$
\begin{aligned}
\theta^2 &= \sup_{x \in \mathcal{X}} \|x\|^2 \left(1 + \frac{2\sigma^2}{\gamma^2}\right)^{-(n-1)/2} \left(\frac{1}{2\pi\sigma^2}\right) \int_0^\infty \int_0^\infty e^{-(u_n - w_n)^2/2\gamma^2} e^{-u_n^2/2\sigma^2} e^{-w_n^2/2\sigma^2} u_n w_n \mathrm{d}u_n \mathrm{d}w_n \\
&\leq \sup_{x \in \mathcal{X}} \|x\|^2 \left(1 + \frac{2\sigma^2}{\gamma^2}\right)^{-(n-1)/2} \left(\frac{1}{2\pi\sigma^2}\right) \int_0^\infty \int_0^\infty e^{-u_n^2/2\sigma^2} e^{-w_n^2/2\sigma^2} u_n w_n \mathrm{d}u_n \mathrm{d}w_n \\
&= \sup_{x \in \mathcal{X}} \|x\|^2 \left(1 + \frac{2\sigma^2}{\gamma^2}\right)^{-(n-1)/2} \left(\frac{1}{2\pi\sigma^2}\right) \left(\int_0^\infty e^{-w_n^2/2\sigma^2} w_n \mathrm{d}w_n\right)^2 \\
&= \sup_{x \in \mathcal{X}} \|x\|^2 \left(1 + \frac{2\sigma^2}{\gamma^2}\right)^{-(n-1)/2} \left(\frac{1}{2\pi\sigma^2}\right) \left(\left(-\sigma^2 e^{-w_n^2/2\sigma^2}\Big|_{w_n=0}^\infty\right)\right)^2 \\
&= \sup_{x \in \mathcal{X}} \|x\|^2 \left(1 + \frac{2\sigma^2}{\gamma^2}\right)^{-(n-1)/2} \left(\frac{1}{2\pi\sigma^2}\right) \sigma^4 \\
&= \sup_{x \in \mathcal{X}} \|x\|^2 \left(1 + \frac{2\sigma^2}{\gamma^2}\right)^{-(n-1)/2} \left(\frac{\sigma^2}{2\pi}\right).
\end{aligned}
$$

$\square$

# D  Theoretical constants of STAR RKHS Weightings

We showed in Section 4 how to transform an RKHS Weighting instantiation into a STAR model, while maintaining the form of the model as an RKHS Weighting (Table 4 contains the details). In particular, all algorithms and theoretical guarantees from Dubé & Marchand (2024) still apply. The guarantees involve the theoretical constants:

$$
\kappa := \sup_{x \in \mathcal{X}} \sqrt{\mathbb{E}_{w \sim p} \left[\|\phi(w, x)\mathcal{K}(w, \cdot)\|_{\mathcal{H}}^2\right]} = \sup_{x \in \mathcal{X}} \sqrt{\mathbb{E}_{w \sim p}[\mathcal{K}(w, w)\phi(w, x)^2]} \tag{57}
$$

$$
\theta := \sup_{x \in \mathcal{X}} \left\|\mathbb{E}_{w \sim p}[\phi(w, x)\mathcal{K}(w, \cdot)]\right\|_{\mathcal{H}} = \sup_{x \in \mathcal{X}} \sqrt{\mathbb{E}_{w \sim p}\mathbb{E}_{u \sim p}[\mathcal{K}(u, w)\phi(u, x)\phi(w, x)]}, \tag{58}
$$

defined for any instantiation $(\mathcal{W}, \phi, \mathcal{K}, p)$.

Here, we explain how to obtain those constants, if desired, for a STAR RKHS Weighting instantiation. For $\kappa$, we have:

$$
\begin{aligned}
\kappa^2 &= \sup_{x \in \mathcal{X}} \mathbb{E}_{(w,I) \sim p_{\mathcal{W}|I} \times p_{[n]}} \left[\mathcal{K}((w, I), (w, I))\phi_I(w, x_I)^2\right] \\
&= \sup_{x \in \mathcal{X}} \mathbb{E}_{I \sim p_{[n]}} \mathbb{E}_{w \sim p_{\mathcal{W}|I}} \left[\mathcal{K}_W(w, w)\mathbb{1}[I = I]\phi_I(w, x_I)^2\right] \\
&= \sup_{x \in \mathcal{X}} \mathbb{E}_{I \sim p_{[n]}} \mathbb{E}_{w \sim p_{\mathcal{W}|I}} \left[\mathcal{K}_W(w, w)\phi_I(w, x_I)^2\right] \\
&\leq \mathbb{E}_{I \sim p_{[n]}} \sup_{x \in \mathcal{X}} \mathbb{E}_{w \sim p_{\mathcal{W}|I}} \left[\mathcal{K}_W(w, w)\phi_I(w, x_I)^2\right]
\end{aligned}
$$

The expression $\sup_{x \in \mathcal{X}} \mathbb{E}_{w \sim p_{\mathcal{W}|I}} \left[\mathcal{K}_W(w, w)\phi_I(w, x_I)^2\right]$ is in fact the constant $\kappa^2$ for the partial instantiation defined only on the feature subset $I$. Let us denote it $\kappa_I^2$. Then we have:

$$
\kappa^2 \leq \mathbb{E}_{I \sim p_{[n]}} \kappa_I^2. \tag{59}
$$

This quantity depends on the distribution of the feature subsets $p_{[n]}$. In the simplest cases, where for example $p_{[n]}$ is the uniform distribution over all feature subsets of size $k$, then the constant $\kappa_I$ might simply be the

same for all $I$. In the worst cases, we can take the supremum of $\kappa_I$ over all possible subsets $I$ instead of the expectation. As for $\theta$:

$$
\begin{aligned}
\theta^2 &= \sup_{x \in \mathcal{X}} \mathbb{E}_{(w,I) \sim p_{\mathcal{W}|I} \times p_{[n]}} \mathbb{E}_{(u,J) \sim p_{\mathcal{W}|J} \times p_{[n]}} [\mathcal{K}((u,J),(w,I))\phi_J(u,x_J)\phi_I(w,x_I)] \\
&= \sup_{x \in \mathcal{X}} \mathbb{E}_{(w,I) \sim p_{\mathcal{W}|I} \times p_{[n]}} \mathbb{E}_{(u,J) \sim p_{\mathcal{W}|J} \times p_{[n]}} [\mathcal{K}_{\mathcal{W}}(u,w)\mathbb{1}[I=J]\phi_J(u,x_J)\phi_I(w,x_I)] \\
&= \sup_{x \in \mathcal{X}} \mathbb{E}_{(w,I) \sim p_{\mathcal{W}|I} \times p_{[n]}} \mathbb{E}_{u \sim p_{\mathcal{W}|I}} [p_{[n]}(I)\mathcal{K}_{\mathcal{W}}(u,w)\phi_I(u,x_I)\phi_I(w,x_I)] \\
&= \sup_{x \in \mathcal{X}} \mathbb{E}_{I \sim p_{[n]}} p_{[n]}(I) \mathbb{E}_{w \sim p_{\mathcal{W}|I}} \mathbb{E}_{u \sim p_{\mathcal{W}|I}} [\mathcal{K}_{\mathcal{W}}(u,w)\phi_I(u,x_I)\phi_I(w,x_I)] \\
&\leq \mathbb{E}_{I \sim p_{[n]}} p_{[n]}(I) \sup_{x \in \mathcal{X}} \mathbb{E}_{w \sim p_{\mathcal{W}|I}} \mathbb{E}_{u \sim p_{\mathcal{W}|I}} [\mathcal{K}_{\mathcal{W}}(u,w)\phi_I(u,x_I)\phi_I(w,x_I)].
\end{aligned}
$$

Exactly as before, the expression $\sup_{x \in \mathcal{X}} \mathbb{E}_{w \sim p_{\mathcal{W}|I}} \mathbb{E}_{u \sim p_{\mathcal{W}|I}} \left[ p_{[n]}(I)\mathcal{K}_{\mathcal{W}}(u,w)\phi_I(u,x_I)\phi_I(w,x_I) \right]$ is the constant $\theta^2$ for the partial model defined on the subset $I$. We can write it as $\theta_I^2$ and:

$$
\theta^2 \leq \mathbb{E}_{I \sim p_{[n]}} p_{[n]}(I)\theta_I^2. \tag{60}
$$

In the event that $\theta_I \leq \hat{\theta}$ for all $I$ for some $\hat{\theta}$, and that $p_{[n]}(I)$ is constant, then we have:

$$
\theta^2 \leq p_{[n]}(I)\hat{\theta}^2, \tag{61}
$$

or:

$$
\theta \leq \sqrt{p_{[n]}(I)}\hat{\theta}. \tag{62}
$$

# E   Details of experimentation

### Preprocessing of the datasets

All datasets have been scaled to have mean 0 and standard deviation 1 on all variables, including the target labels. Means and standard variations were calculated on the training data, then the transformation applied to both training and test datasets.

### Numerical stability of the learning algorithm

To learn RKHS Weightings, we used Algorithm 3 of Dubé & Marchand (2024), the least squares fit of the coefficients. However, we have run into some rare numerical instability problems which led to failure of learning and even negative training $R^2$ scores, which is impossible in theory (at worst, the algorithm could output $0 \in \mathcal{H}$, which has an $R^2$ of 0 exactly). To solve this issue, we added a very small $\ell^2$ regularizer to the optimization objective (Equation 53 of Dubé & Marchand (2024)), which is now:

$$
\mathcal{L}_{\mathcal{S}}^{\mathrm{reg}}(\Lambda\alpha(x)) = \frac{1}{2m}\|\Phi a - \mathbf{y}\|_2^2 + \frac{\lambda}{2}a^\top G a + \frac{\epsilon}{2}a^\top I a, \tag{63}
$$

where $I$ is the identity matrix, and $\epsilon$ is the $\ell^2$ regularization parameter, which we have set to $10^{-10}$. The minimizer $a$ of the previous expression is the solution to the linear problem:

$$
\left(\Phi^\top\Phi + m\lambda G + m\epsilon I\right)a = \Phi^\top\mathbf{y}. \tag{64}
$$

### Hyperparameter selection

As pointed out in Dubé & Marchand (2024), instantiation RWSign, and now also RWRelu, exhibits exponential behavior in the dimensionality $n$ of the instance space. This can immediately be seen in the $c := \left(1 + \frac{\sigma^2}{\gamma^2}\right)^{-n/2}$ coefficient in Equation (25), shared by both RWSign and RWRelu. Instead of using $\sigma$

| | Cross-validation parameters | Source code (clickable) |
|---|---|---|
| DecisionTreeRegressor | max depth $\in \{2, 5, 10, 20\}$ | Scikit-learn |
| EBM | max bins $\in \{512, 1024, 2048\}$
learning rate $\in \{512, 1024, 2048\}$
max rounds $\in \{15000, 25000, 35000\}$
min samples leaf $\in \{1, 2, 3\}$ | InterpretML |
| KernelRidge | kernel = rbf
alpha $\in \{0.01, 0.05, 0.1, 0.5, 1, 5\}$ | Scikit-learn |
| LinearRegression | | Scikit-learn |
| SVR | C $\in \{0.5, 1, 5, 10, 50\}$ | Scikit-learn |
| RWSign | max theta $\in \{0.1, 0.5, 0.9\}$
$\lambda \in \{10^{-9}, 10^{-8}, 10^{-7}, 10^{-6}, 10^{-5}, 10^{-4}, 10^{-3}, 10^{-2}\}$ | Dubé & Marchand (2024) |
| RWRelu | max theta $\in \{0.1, 0.5, 0.9\}$
$\lambda \in \{10^{-9}, 10^{-8}, 10^{-7}, 10^{-6}, 10^{-5}, 10^{-4}, 10^{-3}, 10^{-2}\}$ | This paper |
| RWStumps | $\sigma \in \{0.01, 0.1, 1\}$
$\gamma \in \{0.01, 0.1, 1\}$
$\lambda \in \{10^{-9}, 10^{-8}, 10^{-7}, 10^{-6}, 10^{-5}, 10^{-4}, 10^{-3}, 10^{-2}\}$ | Dubé & Marchand (2024) |

Table 9: Algorithms and models used in this paper and their hyperparameters.

and $\gamma$ as hyperparameters of the model, they use $\sigma$ and $c$, and calculate $\gamma$ from $\sigma$ and $c$. This way, the effect of dimensionality is held constant regardless of $n$.

We modify this methodology slightly. We instead use $c^2 := \left(1 + \frac{2\sigma^2}{\gamma^2}\right)^{-n/2}$. This is the upper bound for $\theta^2$ calculated by Dubé & Marchand (2024) for RWSign. It can also be found in Lemma C.2, and almost as is in the upper bound for $\theta$ for RWRelu (Lemma C.4). Given $\sigma$ and $c \in (0, 1)$, we get:

$$\gamma^2 = \frac{2}{\sigma^2(c^{-4/n} - 1)}. \tag{65}$$

The advantage of this method is that the model output $\Lambda\alpha(x)$ being upper bounded by $\theta\|\alpha\|_{\mathcal{H}} \leq c\|\alpha\|_{\mathcal{H}}$, this gives meaning to the parameter $c$. For RWSign, this allows setting the maximum value of the model. For RWRelu, the relationship is functionally the same, though the upper bound for $\theta$ is slightly more complex.

Finally, Table 9 contains the hyperparameters of all the algorithms and models used in this paper.

**Reproducing the results**

The code for this paper can be found at `https://github.com/gadub44/star-rkhs-weightings`. The repository contains a `requirements.txt` file containing the particular Python packages that were used in our experiments. They can be installed from the command line with the command `pip install -r requirements.txt`. Table 10 lists the commands to run the experiments of Section 5, and the files they generate.

| | Command | Files generated |
|---|---|---|
| Fig. 1 | `python -m major_experiments.shapley_time` | `results/shapley_time.csv`
`figures/shapley_time_left.pdf`
`figures/shapley_time_right.pdf` |
| Fig. 2 | `python -m major_experiments.shapley_accuracy` | `results/shapley_accuracy-n10-T100.csv`
`results/shapley_accuracy-n100-T100.csv`
`results/shapley_accuracy-n500-T100.csv`
`figures/shapley_accuracy-n10-T100.pdf`
`figures/shapley_accuracy-n100-T100.pdf`
`figures/shapley_accuracy-n500-T100.pdf` |
| Table 6 | `python -m major_experiments.regression --final` | `results/regression-final.csv`
`tables/regression-final.tex`
`tables/regression-final-dense.tex` |
| Table 8 | `python -m major_experiments.time_series --final` | `results/regression-final.csv`
`tables/time-series-final.tex`
`tables/time-series-final-dense.tex` |
| Fig. 3, 4 | `python -m major_experiments.shapley_comparison` | `results/ebm_explanation.pkl`
`results/rkhs_explanation.pkl`
`figures/beeswarm_plot_ebm.pdf`
`figures/beeswarm_plot_rkhs.pdf`
`figures/waterfall_plot_ebm.pdf`
`figures/waterfall_plot_rkhs.pdf` |

Table 10: Commands to run the various experiments in this paper.

