# OpenReview forum: "Shapley Values of Structured Additive Regression Models and Application to RKHS Weightings of Functions"
_TMLR — Accepted by TMLR_

### Review · Reviewer_7iuM · 2024-11-05

**Summary Of Contributions:**

The paper proposes the "STAR-SHAP" algorithm for computing shapely values for "Structured Additive Regression Models" (STAR models).

As the authors note in the paper, STAR models are already known in the literature under different names, such as GAMs with arbitrary interactions or functional decompositions.

The STAR-SHAP algorithm is a model-specific algorithm. It assumes that we can access the model's different feature-wise component functions. The Shapley values are then given by the expectation of a weighted sum involving the component functions (equation (16) in the paper).

The advantage of the STAR-SHAP algorithm is that it does not involve a sum over exponentially many terms (as in the Shapley formula) but only over the terms present in the model.

The paper also shows that particular classes of models from Dubé & Marchand (2024) can be cast as STAR models.

The paper has several empirical applications, all regression experiments.

**Audience:**

Yes

**Broader Impact Concerns:**

Theory / Algorithms paper; broader impact statement seems uneccesary.

**Claims And Evidence:**

Yes

**Requested Changes:**

- The paper needs to discuss/compare with other more recent algorithms to compute shapley values for black-box models. For example: https://proceedings.neurips.cc/paper_files/paper/2023/hash/264f2e10479c9370972847e96107db7f-Abstract-Conference.html

- it would be great if the paper could include a table of different model-specific and model agnostic algorithms in the literature and state the computational complexity of the various algorithms. This would help the reader to situate the paper's result in the literature.

- I would appreciate if the paper could elaborate a bit about the con's and pro's of model-specific / model-agnostic shapley values computation.

- The paper should include examples of Shapley values of STAR models estimated with different methods: the STAR-SHAP algorithm, Kernel SHAP, and perhaps other model-agnostic approximation algorithms?

- Because you use EBMs, you should cite the interpret software package https://arxiv.org/pdf/1909.09223


**Questions:**

**Q1:** What happened to the value function? Does the STAR-SHAP algorithm compute "interventional" shapley values, or does that depend on the type of the expectation operator in equation (16)? I'm asking this because some forms of shapley values (with marginal distribution) are usually much harder to estimate, and model-specific algorithms like treeSHAP usually implicitly rely on a specific value function. It would be great if the authors could clarify this in the paper.

**Q2:** The connection between the RKHS methods and STAR models seems interesting, but to me, it also seems somewhat unmotivated. Why consider this particular, somewhat unusual class of models? How is this connected to the STAR-SHAP algorithm? Is the main reason that you need a method to obtain non-trivial STAR-models?

**Q3:** Perhaps you could briefly explain the connections/differences between equation (16) in your paper and equation (11) in Bordt & von Luxburg (2023) (which appears in many other papers, too).

**Strengths And Weaknesses:**

**Strengths:**

The paper is very clearly written. The paper is also appropriately positioned with respect to the related literature, with the exception of some recent developments in estimating shapley values (see below).

Theorem 3.1 and the proposed algorithm could potentially be interesting to some readers of TMLR.

Generally, reading this paper could be interesting as an introduction for researchers who are not that well-aware of the connections between Shapley Values and functional decompositions.

**Weaknesses:**

The proposed algorithm is a simple consequence of the structure of the model. I don't think the shapley-estimation part of this paper contains much novelty for researchers who are aware of the connections between Shapley Values and functional decompositions.

The paper contains no real performance comparison with other methods that estimate shapley values, except for SHAP. For a paper that introduces a new algorithm that is performant, I would expect a somewhat more extensive performance comparison with other algorithms.

In the experiments, it seems that EBMs perform pretty well, which makes sense because the EBM software package is fairly optimized for the application scenario considered in this paper. So why do we need the newly proposed method?

---

> ### Author Response · Authors · 2024-11-28
> **Response to the reviewer**
>
> We thank the reviewer for the thorough and detailed review. We have submitted a new version of the manuscript which we hope addresses the following issues:
>
> We significantly expanded the section about Shapley values (Section 2.2) to include the original definition (Equation (13) of Shapley et al., 1953), and explain the value function that we use: for a given model $h(x)$ and instance $x$, the value of a feature subset $I$ is $v(I) := E_{z \sim S}[h(r_I(z, x))]$, where $r_I$ is the replacement function which we define in Equation (11), and where $S$ is the background dataset. This is an interventional Shapley value.
>
> Concerning the connection between RKHS Weightings and STAR-SHAP. We mean this paper to make progress on developing the theory and practical side of the new and scarcely-studied family of functions that are RKHS Weightings. The link between RKHS Weightings and STAR-SHAP is simply that we showed that RKHS Weightings can be STAR models, and STAR-SHAP is required to calculate their Shapley values. We added text to clarify this (e.g. the second to last paragraph of the introduction of the revised manuscript).
>
> Concerning Equation (11) of Bordt & von Luxburg (2023), and comparing to other algorithms. The expansion of the Shapley value section also includes a more thorough exposition of other algorithms and work on the subject of Shapley values, including KernelSHAP, Unbiased KernelSHAP and SHAP-IQ. We were not aware of SHAP-IQ (Fumagalli et al., 2023), and thank the reviewer for pointing out this meaningful work. We also added these algorithms to the time experiment (Figure 1), which now more clearly shows the advantage of STAR-SHAP over other algorithms. We also added a new experiment (Figure 2) which assesses the quality of the approximate values returned by SHAP-IQ. There, we also see that using our exact algorithm is beneficial. We also added a table of the various algorithmic complexities (Table 1 in the new manuscript). (However, we are not yet certain of those complexities; our experiments do not seem to corroborate the values we found in the literature.)
>
> Concerning the pertinence of STAR RKHS Weightings in light of EBM’s performance. The experiments in this paper are a proof of concept rather than an optimized application of the RKHS Weighting models, which will require more research. We added text to clarify this (e.g. the last paragraph of Section 5.4, the discussion in Section 5.6).
>
> The expanded Shapley value section also elaborates on model-specific and model-agnostic algorithms.
>
> We cited the EBM software package.
>
> We hope that the reviewer’s concerns have been addressed, and will gladly answer any further question.

---

> > ### Comment · Reviewer_7iuM · 2024-11-29
> > **Response to the authors**
> >
> > Thanks to the authors for addressing many points in the revised manuscript.
> >
> > I specifically like Table 1. Concerning your comment that you are not certain about the complexities in Table 1, and that the experiments do not seem to corroborate the values, it would be great if you could investigate this a bit further and also comment on any observed discrepancies in the final manuscript.
> >
> > As a final comment, I would suggest to re-write the first three sentences of the abstract.

---

> > > ### Author Response · Authors · 2024-12-05
> > > **Response to the reviewer**
> > >
> > > We submitted a new version with a revised abstract. We also solved the confusion about the complexities in Table. In short:
> > >
> > > 1- The software package that we used (SHAP-IQ) implements its own faster (but equivalent) version of Unbiased KernelSHAP, which explains that the complexity of Unbiased KernelSHAP in our experiments does not correspond to that of its original implementation.
> > >
> > > 2- The SHAP-IQ paper states that its complexity is linear in the dimensionality $n$. However, that is for the computation of a single Shapley value. Computing all $n$ Shapley values is therefore quadratic, as we see in Figure 1.
> > >
> > > 3- We verified that KernelSHAP is cubic in $n$, as we state in Table 1. It simply appears linear at the scale of Figure 1b.
> > >
> > > These clarifications are in the captions of Table 1 and Figure 1.

---

> > > > ### Comment · Reviewer_7iuM · 2024-12-05
> > > > **Reply to the authors**
> > > >
> > > > Thanks for the additional clarifications.

---

### Review · Reviewer_au4X · 2024-11-11

**Summary Of Contributions:**

This paper proposed an efficient method for computing Shapley values of structured additive regression (STAR) models. Additionally, the authors introduced a novel instantiation of RKHS Weighting that was better suited for regressions, transforming it to fit within the STAR model framework.

**Audience:**

Yes

**Broader Impact Concerns:**

See weaknesses.

**Claims And Evidence:**

Yes

**Requested Changes:**

No concerns.

**Strengths And Weaknesses:**

**Strengths:**

1.	This paper proposed an algorithm for computing Shapley values for a family of models, i.e., structured additive regression models, which were the generalization of generalized additive models (GAMs).

2.	The authors presented an instantiation of RKHS Weightings and showed how to transform an RKHS Weighting model into a STAR model.

**Weaknesses:**

1.	This paper should explicitly compare the time complexity of approximating Shapley values for all models versus STAR models, i.e., the time complexity of Algorithm 1. To highlight the contribution, the authors could list commonly used STAR models below Eq. (11) to clarify the application scenarios for this algorithm. Additionally, a table comparing typical use cases for Shapley values with RKHS-weighted models, STAR models, and all models would be helpful, along with the computational costs for full n-way interactions and k-way interactions (where $k \le n$).

2.	The purpose of introducing RKHS weighting in Eq. (12) is unclear. Since the authors referenced an unpublished paper [Dub. & Marchand (2024)], it would be beneficial to provide background on RKHS, RWSign, and RWStumps. Specifically, the authors should clarify why RKHS weighting is needed in the context of Shapley values for STAR models, the model family it represents, the typical models it includes, and its potential applications. Additionally, the authors should explain the fundamental differences between RWSign and RWStumps, and specify the scenarios for which each method is applicable.

3.	Algorithm 1 did not essentially reduce the exponential complexity of Shapley values. Instead, it reduced the computational cost by constraining the number of interacting variables in the model. The authors should note this in the introduction. Experimentally, the authors should evaluate models with interactions involving more than just 5 variables in Figure 1. For example, in Section 6, they could include models with 1 to 10-way interactions, and report the model’s performance, providing an illustration of the algorithm’s limitations.

---

> ### Author Response · Authors · 2024-11-28
> **Response to the reviewer**
>
> We thank the reviewer for their review. We have submitted a new version of the manuscript which we hope addresses some of the reviewer's concerns.
>
> Concerning a comparison of various Shapley value algorithms and models. We added a section (3.1) which specifically compares STAR-SHAP to other algorithms. The section contains a table listing various Shapley value exact calculation and approximation algorithms and their target model families, along with their computational complexity (Table 1 in the new manuscript; however, we are not yet certain of those complexities, as our experiments do not seem to corroborate the values we found in the literature. This will be fixed for the final version of the manuscript.). We also added these Shapley value algorithms to Figure 1 to more fully compare STAR-SHAP to existing algorithms. We also added a short section (3.2) describing models the algorithm can be applied to.
>
> We have expanded the RKHS Weighting section (2.3), and hope that it more clearly defines the model family. As for their inclusion in this paper, we added text to clarify this (e.g. the second to last paragraph of the introduction). In short, the paper seeks to advance the understanding and applicability of this newly-proposed family, which we did by casting the model as STAR models. The purpose of STAR-SHAP is then to interpret those models, but has the added benefit of working on many more model classes.
>
> We made clearer in the introduction that the algorithm we present scales exponentially in the maximal size of the interactions, and thus does not solve the exponential complexity of calculating the exact Shapley values for arbitrary models. As for illustrating that fact in Figure 1, the red line (STAR-SHAP (n-STAR)) shows the exponentially increasing cost of computing the Shapley values for a model using n-way interactions for up to n=10, where the computational cost started to become prohibitive. We also added a Figure 2 which compares STAR-SHAP to a Shapley value approximation algorithm for various values of $k$, the number of variable interactions in the model. The text also now analyses both Figures in more detail.
>
> We will happily provide any more clarification that the reviewer desires.

---

> > ### Comment · Reviewer_au4X · 2024-11-29
> >
> > Thank you for your detailed response. The addition of Table 1 and the experimental results in Figures 1 and 2 has significantly improved the completeness and clarity of the paper. Furthermore, the revisions to the Introduction and the introduction of RKHS Weighting have strengthened the overall rigor of the work. I have no further questions.

---

### Review · Reviewer_nmfe · 2024-11-15

**Summary Of Contributions:**

The purpose of the paper is twofold: the authors find a closed form expression to compute the Shapley values of a family of models, called Structured Additive Regression (STAR) models, and presents an algorithm for their calculation. This algorithm allows to compute in a reasonable amount of time the Shapley values, if the maximal number of variable interactions is small. The major part of the paper is devoted to providing connections between RKHS models and STAR models. Finally, the paper is completed with applications on various datasets.

**Audience:**

Yes

**Broader Impact Concerns:**

It is not clear from the article to what extent the proposed models are actually used and considered relevant in practical applications.

**Claims And Evidence:**

No

**Requested Changes:**

1. The paper *Khorrami Chokami, A., \& Rabitti, G. (2024). An Exact Game-Theoretic Variable Importance Index for Generalized Additive Models. Journal of Computational and Graphical Statistics, 1–10. DOI: 10.1080/10618600.2024.2327577* provides closed-form results for Shapley values for GAMs, along with confidence intervals. Please clarify and contrast your contribution (for GAMs) with respect to that paper.

2. Concerning the application, consider to add the computed Shapley values to interpret the results. This would improve the overall interpretation and relevance of the findings.

3. Cite *Owen, A. B. (2014), Sobol’ Indices and Shapley Value, SIAM/ASA Journal on Uncertainty Quantification, 2, 245–251. DOI: 10.1137/130936233* and *Wood, S. N., Goude, Y., and Shaw, S. (2015), Generalized Additive Models for Large Data Sets, Journal of the Royal Statistical Society, Series C, 64, 139–155. DOI: 10.1111/rssc.12068*}.

**Strengths And Weaknesses:**

I think the article requires significant revisions to meet academic standards, and I propose a major revision for the following motivations.

1. The text contains unclear sentences and at times employs overly simplistic or even confusing English. For example, after Equation (6), the authors write [...] the set of variables in $\pi$, but $\pi$ is a function to create permutations and it is not a set. I also understand that $U(\Omega)$ is the Uniform distribution, but this should be stated explicitly. Also, if $S$ is a dataset and not a random variable, what does it mean $z\sim S$ in the expression of the expected values? Moreover, the use of contractions is not suitable for the academic tone required for this publication.
2. The language and explanations are not accessible to readers who are not already experts in the subject. The number of acronyms present in the manuscript is excessive and distracts the reader. For example, the terms *RKHS, ReLU* (even if their meaning can be guessed), are introduced without explanation.
3. The section on Shapley values lacks sufficient detail and fails to clarify fundamental aspects.

    a. The value function used in the Shapley value calculations is not explicitly defined or explained. An explicit description of this function and its role in interpreting Shapley values is necessary.

    b. The text does not adequately discuss the Shapley values in the practical applications. The datasets are presented only in terms of sample dimension. Basically, the applications are presented as: "These are the datasets, these are the $R^2$ of the various models, in Figure 1 there is the computational cost of the Shapley values of a Structured Additive Regression model with respect to the dimensionality of the dataset (which one?)." I really find the part on the Shapley values quite poor, but if I look at the title and the abstract I expect this to be the core of the paper.

    c. Page 5 - line 12 from the bottom: *[...] then this formula can be calculated in $O(mN 2^k)$, independently of the dimensionality of the data.* Since $m=|S|$ where $S$ is the dataset and $N$ is the number of features which satisfy a certain condition (with $k$ as maximum number), it is unclear why the complexity is independent of data dimensionality. Not only, is it really necessary to introduce $m$ as a new notation?

---

> ### Author Response · Authors · 2024-11-28
> **Response to the reviewer**
>
> We thank the reviewer for their time reviewing our paper. We have submitted a new version of the manuscript with substantial modifications and additions. Pertaining specifically to the reviewer’s concerns:
>
> We have corrected the language and notation as requested. We will gladly correct any remaining issue the reviewer observes.
>
> We used the full name for RKHS and ReLU on their first usage, and gave the mathematical definition of the ReLU, and an informal definition of RKHS. (We think that giving the formal mathematical definition of reproducing kernel Hilbert spaces is unnecessary, as it is not used in the paper). We also improved the entire section defining RKHS Weightings (2.3).
>
> We made significant additions to the Shapley value section (2.2). It now includes the original form of the Shapley value (from the 1953 paper), and the value function (Eq. (12)) which gives rise to our version of the Shapley value (Eq. (14)).
>
> The computational complexity of our algorithm does not explicitly depend on the dimensionality of the input space. The reason for this is the fact that the algorithm loop only requires calculating the output of the partial functions $f_I(x_I)$ (see Eq. (20)). The cost of this operation depends only on the size of the feature subset $I$, independently of the total number of variables. This is why the computational complexity depends only on the maximal size $k$ of the feature subsets. We added this justification for the algorithmic complexity (the last paragraph on page 7 of the new manuscript). We also changed the form of the algorithmic complexity slightly.
>
> We added an experiment (Section 5.5 of the new manuscript) which compares the Shapley values of a STAR RKHS Weighting model and of an Explainable Boosting Machine model, both trained on the same dataset. We also added a discussion afterward in Section 5.6 to summarize the conclusions of the various experiments. We do not claim RKHS Weightings should be used over other models at this point, only that they show promise and should be considered as a plausible model.
>
> We address Khorrami Chokami, A., & Rabitti, G. (2024) in the final paragraph of Section 2.2.
>
> We cited Wood, S. N., Goude, Y., and Shaw, S. (2015) where it was appropriate.
>
> Finally, while the presence of many acronyms is unfortunate, it is somewhat unavoidable, unless the reviewer can propose a solution. For example, would a table listing all acronyms used in the paper be of help?
>
> We will gladly answer any further question.

---

> > ### Comment · Reviewer_nmfe · 2024-12-28
> >
> > Thank you for the detailed responses; I am satisfied with the current state of the paper. Regarding your proposed solution, 'For example, would a table listing all acronyms used in the paper be of help?', I would say this is the best approach. I have no further questions or requests to make.

---

> > > ### Author Response · Authors · 2025-01-06
> > >
> > > Thank you for the reply. We submitted a new version of the manuscript with a table of acronyms on page 2.

---

### Decision · Action_Editor_hqL8 · 2025-01-03

**Recommendation:** Accept with minor revision

**Comment:**

All reviewers recommended acceptance (at least weakly) after the rebuttal and all indicated explicitly that their concerns were addressed.

I do note that Reviewer nmfe suggested adding a table of acronyms, which the authors should consider for the camera-ready revision.

**Audience:**

The submission is of interest because it contributes to efficient calculation of Shapley values for feature attribution/explanation.

**Claims And Evidence:**

This submission has two main contributions:
1. An efficient algorithm for calculating Shapley feature importance values for Structured Additive Regression (STAR) models;
1. A novel instantiation of a recently introduced RKHS Weightings of Functions paradigm, which can be transformed into a corresponding class of STAR models and thus explained using contribution 1.

Regarding the first contribution, reviewers had concerns mainly about experimental comparisons with additional Shapley value algorithms and better situating the contribution in the context of model-specific and model-agnostic Shapley value algorithms. The authors addressed these points during the discussion period, as summarized by the additions of/to Table 1, Figure 1, and Figure 2.

Regarding the second contribution, reviewers questioned the motivation for the new class of models. The authors responded that the purpose is to advance the understanding and viability of RKHS Weightings models, by showing that some of them are STAR models and thus efficiently explainable.

Reviewers also had concerns about unclear notation and other aspects, notably the value function underlying the Shapley values. These were also addressed.